# Efficient Block Bi-clustering by Alternating Semidefinite Programming Relaxation

**Yuxin Ma**  *yuxin.ma@matfyz.cuni.cz*
*Department of Numerical Mathematics*
*Charles University*

**Weiguo Gao**  *wggao@fudan.edu.cn*
*School of Mathematical Sciences and Shanghai Key Laboratory of Contemporary Applied Mathematics*
*Fudan University*

**Xiang Zhou**  *xizhou@cityu.edu.hk*
*Department of Mathematics*
*City University of Hong Kong*

**Reviewed on OpenReview:** *https://openreview.net/forum?id=xPA4Xg0IvL*

## Abstract

The bi-clustering problem is a common task in data mining, often formulated as a challenging non-convex optimization problem. In this paper, we address the block bi-clustering problem using a novel formulation with semi-definite programming (SDP) relaxation and two low-rank matrix approximation. Our method alternates between optimizing the row and column membership matrices in a sequential manner, freezing one matrix while solving the subproblem for the other in each step. We prove that the numerical membership matrices generated by our algorithm achieve an error in the Frobenius norm bounded by $O(1/\sqrt{n})$ and $O(1/\sqrt{m})$, ensuring accuracy and scalability as the data dimensions grow. Through experiments on both simulated and real datasets, we demonstrate that our algorithm performs comparably or better than existing bi-clustering methods in terms of both accuracy and efficiency.

## 1 Introduction

Bi-clustering, also known as co-clustering or two-way clustering, is an unsupervised machine learning method that has attracted significant interest across diverse fields such as bioinformatics, text mining, and recommendation systems José-García et al. (2023); Tanay et al. (2005). Unlike traditional clustering, which clusters either rows or columns, bi-clustering involves the joint clustering of rows and columns within a data matrix. The goal of bi-clustering algorithms is to identify subgroups of rows and columns that demonstrate a high level of similarity within a submatrix. This approach is especially valuable in bioinformatics for gene expression data analysis, where it seeks to detect groups of genes displaying similar expression patterns under specific conditions.

Numerous bi-clustering algorithms have been developed, primarily for gene expression data analysis Cheng & Church (2000); Getz et al. (2000); Maâtouk et al. (2021); Madeira & Oliveira (2004); McLachlan et al. (2005); Pontes et al. (2015); Tan & Witten (2014). For example, Cheng and Church Cheng & Church (2000) proposed a bi-clustering method based on the mean squared residue score, while Getz et al. Getz et al. (2000) introduced coupled two-way clustering for gene microarray analysis. Subsequently, many bi-clustering algorithms have emerged rapidly; comprehensive reviews and comparisons can be found in Padilha & Campello (2017). However, identifying biclusters in large-scale datasets is known to be an NP-hard problem, leading to the widespread use of heuristic approaches in practical applications José-García et al. (2023). Despite of the practical good performance of various bi-clustering algorithms on different datasets,

most of the existing bi-clustering algorithms lack the theoretical guarantee to yield a correct bi-clustering under certain assumptions on the input data set, since most models for bi-clustering are typically formulated as intractable combinatorial optimization problems.

Block bi-clustering is a special form of bi-clustering where the data matrix is partitioned into mutually exclusive and exhaustive blocks or submatrices. In this model, each row or column is assigned solely to one specific row or column group, which contrasts with overlapping bi-clustering models where rows or columns can belong to several groups or none. Each block, known as a bicluster, comprises a selection of rows and columns, with the data values displaying a similar pattern. The checkerboard pattern in bi-clustering indicates these non-overlapping blocks.

To address clustering problems, the semidefinite programming (SDP) relaxation is a powerful technique used in approximating combinatorial optimization problems Ames (2014); Li et al. (2021); Mixon et al. (2017); Peng & Wei (2007). SDP is a generalization of linear programming where we optimize a linear objective function subject to the constraint that an affine combination of symmetric matrices is positive semidefinite. An advantage of employing SDP is that it can leverage the power of convex optimization to obtain theoretical guarantees.

Only a few works, such as Ames (2014); Sudoso (2024), explore the algorithmic and theoretical aspects of SDP relaxation for bi-clustering problem. In Ames (2014), the authors employ the alternate direction method of multipliers (ADMM) to solve the relaxed SDP, aiming to identify the densest disjoint $k$ biclusters. Their analysis demonstrates that accurate matrix partitions can be recovered when the underlying biclusters are sufficiently distinct. However, their general SDP formulation is computationally intensive, limiting its practical applicability. Meanwhile, Sudoso (2024) focuses on leveraging SDP relaxation to develop a tailored branch-and-cut algorithm for bi-clustering.

In this work, we formulate the block bicluster problem as the optimization problem for the row and column membership matrices, and apply the SDP relaxation and the low-rank approximation to construct our alternative SDP relaxation algorithm. Our main contributions can be summarized as follows.

- We propose a tuning-free non-convex SDP relaxation for the bi-clustering problem, applicable when the number of row and column clusters is known.

- Our algorithm alternates between clustering rows and columns by solving a convex SDP relaxation in each step, efficiently identifying the checkerboard structure of the data matrix.

- Our method employs low-rank approximation specifically for the membership matrices, ensuring that the clustering structure is captured effectively while maintaining computational efficiency.

- On the theoretical side, for data matrices with Gaussian or Bernoulli noise, we prove that the numerical membership matrices generated by our algorithm achieve an error bound in the Frobenius norm of $O(1/\sqrt{n})$ and $O(1/\sqrt{m})$, ensuring accuracy and scalability as the data dimensions grow.

- Through the simulation data and the empirical lung cancer dataset, we demonstrate that our algorithm remains robust and accurately identifies the checkerboard structure, even under large noise levels.

The remainder of this paper is organized as follows. In Section 2, we describe the model for block bi-clustering problem, and derive the SDP. Then in Section 3, we develop the alternating SDP relaxation algorithm for solving the block bi-clustering problem. The error analysis is shown in Section 4, and Section 5 contains numerical experiments.

## 2 SDP relaxation for bi-clustering problem

### 2.1 Model for bi-clustering problem

We consider the bi-clustering problem within a block model framework, where each observation is represented by an $n \times m$ matrix $X$. We assume $X$ admits a decomposition $X = M + N$, where $M = \mathbb{E}[X]$ is the signal

matrix and $N$ represents a noise matrix with independent, zero-mean random entries. For the ground truth matrix $M$, we assume that it exhibits a checkerboard pattern after applying specific row and column permutations defined by the partitions $\{C_k\}, \{E_l\}$ respectively:

$$\{1, \ldots, n\} = \bigcup_{k=1}^{K_R} C_k, \quad \{1, \ldots, m\} = \bigcup_{l=1}^{K_C} E_l. \tag{1}$$

We assume that $M$ is piece-wise constant w.r.t. $\{C_k\}, \{E_l\}$, i.e.,

$$M_{ij} = c_{k,l} \in (0, 1), \quad \forall i \in C_k, j \in E_l. \tag{2}$$

Let $n_k := |C_k|$ for $1 \leq k \leq K_R$, denote the number of elements in the index set $C_k$. We define the following $n$-by-$n$ *row membership matrix* to encode the assignments of rows:

$$(U^\star)_{ij} = \begin{cases} \frac{1}{n_k}, & i, j \in C_k \text{ for some } k \\ 0, & \text{otherwise}, \end{cases} \tag{3}$$

which can be interpreted as a transition matrix for the random walk that assigns equal probability to all states within the same block $C_k$. Equivalently, if $U^\star$ is known, the partition $\{C_k\}$ can be directly recovered by identifying the row and column indices of the nonzero entries of $U^\star$.

Likewise, by considering the $m$ columns with $K_C$ clusters, we have the $m \times m$ *column membership matrix*

$$(V^\star)_{ij} = \begin{cases} \frac{1}{m_l}, & i, j \in E_l \text{ for some } l \\ 0, & \text{otherwise}, \end{cases} \tag{4}$$

with $m_l := |E_l|$, $l = 1, 2, \ldots, K_C$.

Note that for the above $U^\star$ and $V^\star$, we have that

$$U^\star M = M, \quad MV^\star = M. \tag{5}$$

Empirically, the signal matrix $M$ can be recovered from the observation data matrix by the membership matrices: $M \approx U^\star X V^\star$ since the noise $N$ has the zero mean. In summary, the decomposition of $X = M + N$ for the ground truth $M$ can be converted to finding the transition matrices $U^\star$ in equation 3 and $V^\star$ in equation 4.

## 2.2 SDP relaxation

The fact $M = U^\star M V^\star$ motivates us to propose an optimization approach to solve the bi-clustering problem, by minimizing the Frobenius norm between the data matrix and the averaged matrix $UXV$, $\min_{U,V} \|UXV - X\|_{\mathsf{F}}^2$, within the appropriate family of the unknowns matrices $U$ and $V$. However, this non-convex optimization is extremely difficult to solve. Even though $(U, V) \mapsto \|UXV - X\|_{\mathsf{F}}^2$ is convex by fixing one variable, it is far from being an semi-definite program form. It is also important to specify the feasible sets for the matrices $U$ and $V$. We shall offer the solutions to these two challenges below by the SDP relaxation and give an SDP formulation for the bi-clustering problem.

First of all we observe that $U^\star$ satisfies $U^\star \succeq 0$ (symmetric and positive definite), $U^\star \geq 0$ (non-negative entries), $U^\star \mathbf{1}_n = \mathbf{1}_n$ where all entries of the column vector $\mathbf{1}_n$ are one, and $U^\star$ has the rank $K_R$. Similar properties hold for $V^\star$ too. But this family of matrices with the given rank $K_R$ is *not* convex. We apply the technique of SDP relaxation, which is the process of relaxing a non-convex optimization problem into a convex SDP problem. This relaxation is done by replacing the hard constraint of rank with a softer convex constraint of trace. For the application to other problems like K-means, refer to Peng & Wei (2007); Tepper et al. (2018); Yan et al. (2018).

Therefore, we define the constraint sets $\mathbb{S}_U$ and $\mathbb{S}_V$ as follows

$$\mathbb{S}_U := \left\{ U \in \mathbb{R}^{n \times n} \mid U \succeq 0,\ U \geq 0,\ U\mathbf{1}_n = \mathbf{1}_n,\ \text{Tr}(U) = K_R \right\}, \tag{6}$$

$$\mathbb{S}_V := \left\{ V \in \mathbb{R}^{m \times m} \mid V \succeq 0,\ V \geq 0,\ V\mathbf{1}_m = \mathbf{1}_m,\ \text{Tr}(V) = K_C \right\}, \tag{7}$$

where "Tr" refer to the trace of a matrix. We numerically look for two matrices $U$ and $V$ to approximate $U^\star$ and $V^\star$, respectively, by minimizing the F-norm between the data matrix $X$ and the averaged matrix $\overline{X} = UXV$, subject to these two sets:

$$\min_{U \in \mathbb{S}_U, V \in \mathbb{S}_V} \|UXV - X\|_{\mathsf{F}}^2. \tag{8}$$

Secondly, we note that $(U^\star)^2 = U^\star$ and $(V^\star)^2 = V^\star$ and propose a *surrogate* objective function as follow. *If we assume that $U^2 = U = U^\top$ and $V^2 = V = V^\top$, then*

$$\|UXV - X\|_{\mathsf{F}}^2 = \text{Tr}(VX^\top U^2 XV) - 2\text{Tr}(X^\top UXV) + \|X\|_{\mathsf{F}}^2 = -\text{Tr}(X^\top UXV) + \|X\|_{\mathsf{F}}^2. \tag{9}$$

Thus, our surrogate objective function is $-\text{Tr}\left(X^\top UXV\right)$. To conclude, our SDP relaxation for the block bi-clustering problem takes the form

$$\max_{U \in \mathbb{S}_U, V \in \mathbb{S}_V} F(U, V) := \text{Tr}\left(X^\top UXV\right), \tag{10}$$

where $\mathbb{S}_U$ and $\mathbb{S}_V$ are defined in equation 6 and equation 7. $K_R$ and $K_C$, the number of row clusters and the column clusters, are two hyper-parameters.

We highlight that even though equation 9 requires the (non-convex) constraint $U^2 = U$ and $V^2 = V$, our feasible sets $\mathbb{S}_U$ and $\mathbb{S}_V$ do not include this requirement. If an optimal solution of equation 8 indeed satisfy $U^2 = U$ and $V^2 = V$, then it is also an optimal solution of equation 8. If $X = M$, we know that $U^\star$ and $V^\star$ are the (global) optimal solution of equation 8 and equation 10, with the optimal objective value being zero and $\|X\|_{\mathsf{F}}^2$, respectively. The advantage of equation 10 is the linearity in either $U$ or $V$. equation 10 is non-convex jointly in $(U, V)$, but when fixing one variable in equation 10, one can have the SDP for the other free variable, while the original problem equation 8 fails to have the SDP as the subproblem. In Section 3, we solve equation 10 by the idea of applying SDP alternatively for each variable.

## 2.3 Low-rank approximation

Directly solving the matrix optimization problem equation 10 by searching over two arbitrary matrices is computationally challenging and expensive. However, we observe that the ground truth matrices $U^\star$ and $V^\star$ have ranks $K_R$ and $K_C$, respectively, which are insignificantly smaller than the dimensions of $U^\star$ and $V^\star$. This low-rank structure enables us to develop a scalable variant of equation 10. Such a low-rank approach forms the basis of an efficient method for solving the clustering problem, as discussed in Kulis et al. (2007).

Specifically, we assume $U = (Y_U)^\top Y_U$ and $V = (Y_V)^\top Y_V$, where $Y_U$ and $Y_V$ have the size $r_U \times n$ and $r_V \times m$, respectively. Here $n \geq r_U \geq K_R$ and $m \geq r_V \geq K_C$. Then the feasible sets $\mathbb{S}_U$ and $\mathbb{S}_V$ are restricted to $\mathbb{D}_U$ and $\mathbb{D}_V$ as follows

$$\mathbb{D}_U := \{ U = Y_U^\top Y_U \in \mathbb{R}^{n \times n} \mid Y_U \in \mathbb{R}^{r_U \times n}, Y_U \geq 0, Y_U^\top Y_U \mathbf{1}_n = \mathbf{1}_n, \text{Tr}\left(Y_U^\top Y_U\right) = K_R \}, \tag{11}$$

$$\mathbb{D}_V := \{ V = Y_V^\top Y_V \in \mathbb{R}^{m \times m} \mid Y_V \in \mathbb{R}^{r_V \times m}, Y_V \geq 0, Y_V^\top Y_V \mathbf{1}_m = \mathbf{1}_m, \text{Tr}\left(Y_V^\top Y_V\right) = K_C \}. \tag{12}$$

Then equation 10 can be further formulated as the following form of finding two matrices $Y_U$ and $Y_V$:

$$\max_{Y_U \in \mathbb{R}^{r_U \times n}, Y_V \in \mathbb{R}^{r_V \times m}} F(Y_U^\top Y_U, Y_V^\top Y_V)$$
$$\text{s.t. } Y_U \geq 0,\ Y_U^\top Y_U \mathbf{1}_n = \mathbf{1}_n,\ \text{Tr}(Y_U^\top Y_U) = K_R, \tag{13}$$
$$Y_V \geq 0,\ Y_V^\top Y_V \mathbf{1}_m = \mathbf{1}_m,\ \text{Tr}(Y_V^\top Y_V) = K_C.$$

We shall see later that this low-rank formulation is more cost-effective than equation 10.

## 3 Algoirthm: alternating SDP for block bi-clustering

The optimization problems formulated in equation 10 and equation 13 are non-convex, since both $U$ and $V$ ($Y_U$ and $Y_V$, respectively) are involved in the objective functions $F$. We propose to alternatively maximize one matrix each time to solve equation 10 and equation 13. In the following, we discuss the method for equation 10 first and the applicability to equation 13 is straightforward.

By freezing one of $U$ or $V$ in equation 10, the maximization of the other is a standard convex SDP as follows:

$$
\begin{aligned}
&\max_U f(U) := \mathrm{Tr}\left(A_V^\top U\right), \quad A_V = XVX^\top \\
&\text{s.t.} \quad U \succeq 0, \quad U \geq 0, \quad U\mathbf{1}_n = \mathbf{1}_n, \quad \mathrm{Tr}(U) = K_R,
\end{aligned}
\tag{14}
$$

and

$$
\begin{aligned}
&\max_V g(V) := \mathrm{Tr}\left(A_U^\top V\right), \quad A_U = X^\top U X \\
&\text{s.t.} \quad V \succeq 0, \quad V \geq 0, \quad V\mathbf{1}_m = \mathbf{1}_m, \quad \mathrm{Tr}(V) = K_C.
\end{aligned}
\tag{15}
$$

In our alternating SDP algorithm, the iterative procedure starts from an initial value of one matrix, namely $V_0$. Then we solve the subproblem equation 14 associated with this $V_0$ and set the maximizer of this subproblem as $U_1$, followed by solving the subproblem equation 15 with $U = U_1$ to obtain $V_1$, and so on. We can select any positive definite matrix as the initial $V_0$, as shown in Section 4.1. For example, one can initialize $V_0$ by any clustering method applied to column vectors, for instance by the $K$-means method. A simpler strategy is to just set $V_0$ as the identity matrix.

Likewise, we apply the same alternative optimization technique to equation 13 in the low-rank setting, by solving the following two SDP subproblems in the low-rank approximation alternatively:

$$
\begin{aligned}
&\max_{Y_U \in \mathbb{R}^{r_U \times n}} f(Y_U^\top Y_U) = \mathrm{Tr}\left(A_V^\top Y_U^\top Y_U\right) \\
&\text{s.t.} \quad Y_U \geq 0, \quad Y_U^\top Y_U \mathbf{1}_n = \mathbf{1}_n, \quad \mathrm{Tr}(Y_U^\top Y_U) = K_R,
\end{aligned}
\tag{16}
$$

and

$$
\begin{aligned}
&\max_{Y_V \in \mathbb{R}^{r_V \times m}} g(Y_V^\top Y_V) = \mathrm{Tr}\left(A_U^\top Y_V^\top Y_V\right) \\
&\text{s.t.} \quad Y_V \geq 0, \quad Y_V^\top Y_V \mathbf{1}_m = \mathbf{1}_m, \quad \mathrm{Tr}(Y_V^\top Y_V) = K_C.
\end{aligned}
\tag{17}
$$

Problems equation 16 and equation 17 can be solved significantly faster than Problems equation 14 and equation 15, since it reduces the number of unknowns, respectively, from $O(n^2)$ to $O(r_U n)$ for solving $U$ with $r_U = \mathrm{const} \cdot K_R$ ($\mathrm{const} \geq 1$) in equation 16, and from $O(m^2)$ to $O(r_V m)$ for solving $V$ with $r_V = \mathrm{const} \cdot K_C$ ($\mathrm{const} \geq 1$) in equation 17. If the true maximizers of each subproblems equation 16 and equation 17 do satisfy the rank no more than $r_U$ and $r_V$, respectively, then the two approaches are equivalent.

This alternating iteration method is summarized in Algorithm 1. In principle, any efficient algorithm for the convex SDP problem can be used here. The detailed numerical methods to solve these SDP subproblems are to be specified in Section 5.

The complexity of Algorithm 1 is $O(K_R m^2 + K_C n^2)$. This follows from the fact that the complexity of a low-rank SDP is $O(kn^2)$ Kulis et al. (2007), where $k$ is the rank and $n$ is the dimension of the input matrix. In contrast, both SparseBC Tan & Witten (2014) and SSVD Lee et al. (2010) have complexity $O(K_R K_C mn)$, while COBRA Chi et al. (2017) incurs a higher cost of $O(m^2 n + mn^2)$ due to its convex optimization formulation over pairwise differences.

## 4 Theoretical properties

We investigate the theoretic properties of our algorithm in this section. Recall that the data matrix is given by $X = M + N \in \mathbb{R}^{n \times m}$. In this section, we assume that the numbers of row clusters and column clusters used in Algorithm 1 are always the ground truth: $K_R = K_R^\star$ and $K_C = K_C^\star$.

---

**Algorithm 1:** Alternating SDP for Block Bi-clustering

---

**Input:** The $n$-by-$m$ data matrix $X$; Positive integers $K_R$ and $K_C$; Parameters: stopping tolerance
 `tol` $> 0$, maximum number of epochs `maxiter`.
**Output:** The row clustering matrix $U$ and the column clustering matrix $V$; The estimated block
 matrix $\overline{X}$; The estimated partitions $C$ and $E$ for rows and columns, respectively.
**Program:**

    1 Initialize $V_0$ by spectral clustering of column vectors or simply by $V_0 = I_m$.

    2 For $p = 1 : $ `maxiter`:

        (a) Fixing $V = V_{p-1}$, solve the SDP problem equation 14 or equation 16 to obtain $U_p$.
        (b) Fixing $U = U_p$, solve the SDP problem equation 15 or equation 17 to obtain $V_p$.
        (c) If the difference between $F(U_p, V_p)$ and $F(U_{p-1}, V_{p-1})$ is smaller than `tol`, stop the loop and
            set $U = U_p$ and $V = V_p$.

    3 Compute $\overline{X} = UXV$.

    4 Perform K-means clustering on the rows and columns of $\overline{X}$ with $K = K_R$ and $K = K_C$,
      respectively, to obtain the partitions $C$ and $E$ as defined in equation 1.

---

Without loss of generality, we can assume $M$ in the following form throughout the paper

$$M := \begin{bmatrix} c_{1,1}\mathbf{1}_{n_1}\mathbf{1}_{m_1}^\top & \cdots & c_{1,K_C}\mathbf{1}_{n_1}\mathbf{1}_{m_{K_C^\star}}^\top \\ \vdots & \ddots & \vdots \\ c_{K_R^\star,1}\mathbf{1}_{n_{K_R^\star}}\mathbf{1}_{m_1}^\top & \cdots & c_{K_R^\star,K_C^\star}\mathbf{1}_{n_{K_R^\star}}\mathbf{1}_{m_{K_C^\star}}^\top \end{bmatrix}, \tag{18}$$

and $U^\star$, $V^\star$ are both block diagonal matrices corresponding to the bi-clustering, respectively:

$$U^\star := \begin{bmatrix} \frac{1}{n_1}\mathbf{1}_{n_1}\mathbf{1}_{n_1}^\top & 0 & \cdots & 0 \\ 0 & \frac{1}{n_2}\mathbf{1}_{n_2}\mathbf{1}_{n_2}^\top & \cdots & 0 \\ \vdots & \vdots & \ddots & \vdots \\ 0 & 0 & \cdots & \frac{1}{n_{K_R^\star}}\mathbf{1}_{n_{K_R^\star}}\mathbf{1}_{n_{K_R^\star}}^\top \end{bmatrix} \tag{19}$$

and

$$V^\star := \begin{bmatrix} \frac{1}{m_1}\mathbf{1}_{m_1}\mathbf{1}_{m_1}^\top & \cdots & 0 \\ \vdots & \ddots & \vdots \\ 0 & \cdots & \frac{1}{m_{K_C^\star}}\mathbf{1}_{m_{K_C^\star}}\mathbf{1}_{m_{K_C^\star}}^\top \end{bmatrix}. \tag{20}$$

In the following, we denote the solutions $\hat{U}$ and $\hat{V}$ as the output of the Alternating SDP algorithm (i.e., Algorithm 1) and will show that they can approximate or equal to the true solutions $U^\star$ and $V^\star$, respectively, in the norm sense.

The first part is to study the case where $X = M$, i.e., there is no noise $N$ in the data. The second part is to generalize the result to the noise case. In this work, we consider two widely used probability models. The first is the Gaussian model where $X = \mathbb{E}X + N$ with

$$N_{ij} \sim \mathcal{N}(0, \sigma^2), \quad \text{i.i.d.} \tag{21}$$

The second is the Bernoulli model

$$X_{ij} \sim \mathcal{B}(M_{ij}), \quad \text{i.i.d,} \tag{22}$$

where $\mathcal{B}(p)$ is the binary distribution with the mean $p \in (0, 1)$.

### 4.1 Analysis of the case without noise: $X = M$

In this case of no noise, we will prove that $\hat{U}$ and $\hat{V}$ generated by Algorithm 1 can exactly converge to the true solutions $U^\star$ and $V^\star$ in one step.

**Theorem 1.** *Suppose that $X = M$ with $M$ defined in equation 18. Write the matrix $M$ in row-wise form as*

$$M^\top = [(M^\top)_1, \ldots, (M^\top)_n].$$

*Let $(U_1, V_1)$ be generated by Algorithm 1 with $K_R = K_R^\star$ and $K_C = K_C^\star$ after one step for solving equation 10 or equation 13 with any given $r_U \geq K_R$ and $r_V \geq K_C$.*

*If the initial $V_0 \succ 0$, then*

$$U_1 = U^\star \quad and \quad V_1 = V^\star, \tag{23}$$

*where $U^\star$ and $V^\star$ are defined in equation 19 and equation 20, respectively.*

**Remark 1.** *Note that $U^\star \in \mathbb{D}_U \subseteq \mathbb{S}_U$ and $V^\star \in \mathbb{D}_V \subseteq \mathbb{S}_V$, then we only need to prove Theorem 1 for the case of equation 10.*

Theorem 1 follows directly from the property established below, which is proved in Appendix A.

**Property 1.** *Assume $X = M$ and write this matrix in column-wise form $M = [M_1, \cdots, M_m]$ or in row-wise form $M^\top = [(M^\top)_1, \ldots, (M^\top)_n]$. Then $U^\star$ is an optimal solution of the sub-problem equation 14 with $K_R = K_R^\star$ for any $n \times n$ matrix $V \succeq 0$, and $V^\star$ is an optimal solution of the sub-problem equation 15 with $K_C = K_C^\star$ for any $m \times m$ matrix $U \succeq 0$. The uniqueness is given in the following.*

1. *If $V$ further satisfies*

$$\left((M^\top)_j - (M^\top)_i\right)^\top V \left((M^\top)_j - (M^\top)_i\right) > 0$$

   *for any $i \in C_k$, $j \in C_l$, $k \neq l$, then $U^\star$ is the unique optimal solution of Problem equation 14.*

2. *If $U$ further satisfies*

$$(M_j - M_i)^\top U (M_j - M_i) > 0 \tag{24}$$

   *for any $i \in E_k$, $j \in E_l$, $k \neq l$, then $V^\star$ is the unique optimal solution of Problem equation 15.*

### 4.2 Analysis of the case with noise: $X = M + N$

This section analyzes the Frobenius norm error between the numerical membership matrices computed by Algorithm 1 and the true solution $(U^\star, V^\star)$ in the presence of iid noise for the data $X = M + N$. Our proof focuses on the case of Gaussian noise, as the analysis for other noise distributions follows a similar derivation; see Remark 5.

#### 4.2.1 One step error

**Theorem 2.** *Suppose that $X = M + N$ with $N_{ij} \sim \mathcal{N}(0, \sigma^2)$. If $U_1$ is the optimal solution of Problem equation 14 with $K_R = K_R^\star$ and $V = I$, or if $U_1 = (Y_U)_1^\top (Y_U)_1$, where $(Y_U)_1$ is the optimal solution of Problem equation 16 with $K_R = K_R^\star$ and $V = I$, then for any $0 < \eta < 1$, with the probability $\geq 1 - \eta$, $U_1$ satisfies:*

$$\|U^\star - U_1\|_\mathsf{F} \leq \alpha_0 \frac{\delta_r}{\sqrt{n}}$$

*with*

$$\Delta_r := \min_{i \neq j} \max_{t=1,\ldots,K_C} |c_{i,t} - c_{j,t}|,$$

$$\delta_r := \frac{\sigma}{\Delta_r^2} \max \left( \max_{k=1,\ldots,K_R; l=1,\ldots,K_C} |c_{k,l}|, \sigma \right),$$

$$\alpha_0 := 4\sqrt{mn} \left( \sqrt{n} + \sqrt{m} + \sqrt{2\ln \frac{2}{\eta}} \right) \left( 3\sqrt{n} + \sqrt{m} + \sqrt{2\ln \frac{2}{\eta}} \right) / (n_{\min} \cdot m_{\min}),$$

$$n_{\min} := \min_{i=1,\ldots,K_R} n_i, \quad m_{\min} := \min_{i=1,\ldots,K_C} m_i.$$

**Remark 2.** *When $m/m_{\min} = \Theta(1)$, $n/n_{\min} = \Theta(1)$, $n/m = \Theta(1)$ and set $\ln(2/\eta) = O(\sqrt{mn})$, then we also have that $\alpha_0 = \Theta(1)$. Thus, Theorem 2 implies that*

$$\|U^\star - U_1\|_{\mathsf{F}} = O\left(\frac{1}{\sqrt{n}}\right).$$

Note that Theorem 2 implies the conclusion in Theorem 1 in the special case of no noise in the data matrix $X$, i.e., $\sigma = 0$. In addition, we also have the similar result for the error of $V$.

*Proof.* See Appendix B. □

### 4.2.2 p-step error

The bound in Theorem 2 is the error for one step if the other variable is frozen as the identity matrix, which typically scales as $O(1/n)$. Now we consider the error after $p$ iterations to investigate the effect of maximizing surrogate loss $F(U, V) = \text{Tr}\left(X^\top U X V\right)$ in equation 10, alternatively.

Let $U_p$ and $V_p$ $(p = 1, 2, \dots)$ denote the sequence of matrices generated by Algorithm 1, with $K_R = K_R^\star$ and $K_C = K_C^\star$, initialized at $V_0 = I$. Specifically, $U_p$ and $V_p$ are the optimal solutions to Problem equation 14 (or equation 16) with $V = V_{p-1}$, and Problem equation 15 (or equation 17) with $U = U_p$, respectively. Then for $p \geq 1$,

$$F(U_p, V_p) \geq F(U_p, V_{p-1}) \geq F(U_{p-1}, V_{p-1}). \tag{25}$$

As a result, $F_p := F(U_p, V_p)$ is a monotonic decreasing sequence as $p$ increases.

Our main result is the following theorem with the proof given in Appendix C.

**Theorem 3.** *Under the same assumptions of Theorem 2, we have that for any $\eta > 0$, with probability $\geq 1 - \eta$,*

$$\|U^\star - U_p\|_{\mathsf{F}} \leq 2\left(\frac{F(U^\star, V^\star) - F(U_p, V_{p-1})}{n_{\min} \cdot m_{\min} \cdot \Delta_r^2} + \frac{\alpha_U}{n}\delta_r\delta_c\right)^{\frac{1}{2}} + \frac{\beta_U}{\sqrt{n}}\delta_r, \qquad \forall p \geq 2 \tag{26}$$

*and*

$$\|V^\star - V_p\|_{\mathsf{F}} \leq 2\left(\frac{F(U^\star, V^\star) - F(U_p, V_p)}{n_{\min} \cdot m_{\min} \cdot \Delta_c^2} + \frac{\alpha_V}{m}\delta_r\delta_c\right)^{\frac{1}{2}} + \frac{\beta_V}{\sqrt{m}}\delta_c, \qquad \forall p \geq 1 \tag{27}$$

*with*

$$\Delta_c := \min_{i \neq j} \max_{t=1,\dots,K_R} |c_{t,i} - c_{t,j}|,$$

$$\delta_c := \frac{\sigma}{\Delta_c^2} \max\left(\max_{k=1,\dots,K_R; l=1,\dots,K_C} |c_{k,l}|, \ \sigma\right),$$

*where $\alpha_U$, $\alpha_V$, $\beta_U$ and $\beta_V$ are functions of $m/m_{\min}$, $n/n_{\min}$, $n/m$, $\eta$, $\delta_r/\delta_c$, and also $\delta_c/\delta_r$.*

**Remark 3.** *If $m/m_{\min} = \Theta(1)$, $n/n_{\min} = \Theta(1)$, $n/m = \Theta(1)$, $\ln(2/\eta) = O(\sqrt{mn})$, $\delta_r/\delta_c = \Theta(1)$, and $\delta_c/\delta_r = \Theta(1)$, then $\alpha_U$, $\alpha_V$, $\beta_U$ and $\beta_V$ are all $\Theta(1)$. Then*

$$\|U^\star - U_p\|_{\mathsf{F}} = O\left(\frac{1}{\sqrt{n}}\right), \quad \|V^\star - V_p\|_{\mathsf{F}} = O\left(\frac{1}{\sqrt{m}}\right).$$

**Remark 4.** *As $p$ increases, both $\text{Tr}(XV^\star X^\top U^\star) - \text{Tr}(XV_p X^\top U_p)$ and $\text{Tr}(XV^\star X^\top U^\star) - \text{Tr}(XV_{p-1}X^\top U_p)$ decrease monotonically, since the sequence $F_p$ is decreasing in $p$. Theorem 3 indicates that $(U_p, V_p)$ get closer to $(U^\star, V^\star)$ as $p$ increases until $\text{Tr}(XV_p X^\top U_p)$ saturates, which justifies the stopping criterion used in Algorithm 1. However, the error terms of $\delta_c$ and $\delta_r$ due to the noise is still $O(1/\sqrt{n})$ regardless of $p$.*

**Remark 5.** *In the above proof of Theorem 3 in Appendix C, to handle the noise matrix $N$, we only need an estimate of $\|N\|_2$ in terms of its size $n$ and $m$, which takes the form equation 53 when $N$ is Gaussian. If we consider the other type of noise, the proof still follows with if the similar estimate is provided. For example, if $N$ follows the Bernoulli distribution, then by changing the estimate for Gaussian noise in (Vershynin, 2012, Corollary 5.35), to that for Bernoulli noise in (Vershynin, 2012, Theorem 5.37), we still have $\|N\|_2 \leq O(\sqrt{n} + \sqrt{m})$ with high probability, then it follows that the conclusion*

$$\|U^\star - U_p\|_{\mathsf{F}} \leq O\left(\frac{1}{\sqrt{n}}\right), \quad \|V^\star - V_p\|_{\mathsf{F}} \leq O\left(\frac{1}{\sqrt{m}}\right)$$

*still holds as in Remark 3.*

## 5 Numerical implementation and results

In this section, we provide numerical implementation details for the SDP solvers used in Algorithm 1 and demonstrate the feasibility and efficiency of the alternating SDP framework for bi-clustering through several illustrative examples.

The practical performance of Algorithm 1 depends on the efficiency of the convex SDP solver for the sub-problems. To solve equation 14 and equation 15 via alternating iteration, we employ the alternating direction dual augmented Lagrangian method Wen et al. (2010), which is robust, efficient, and provably convergent; see Appendix D for a brief review and implementation details. For large-scale problems with very large $n$ and $m$, we recommend the low-rank SDP technique to solve equation 16 and equation 17. To this end, we adopt the low-rank SDP framework from Kulis et al. (2007) for the subproblems; its implementation is described in Appendix E. This low-rank SDP solver is significantly faster, much more scalable, and often yields better numerical results compared to applying Wen et al. (2010) directly.

To recover the partitions $C$ and $E$ defined in equation 1 for the bi-clustering problem, we apply $K$-means clustering to the rows and columns of $\bar{X}$ with $K = K_R$ and $K = K_C$, respectively. The resulting cluster assignments yield the estimated partitions $\hat{C}$ and $\hat{E}$, which are then compared with the outputs of other bi-clustering methods for the clustering accuracy. To minimize the impact of initialization in $K$-means method, we repeat ten times of $K$-means and report the averaged clustering accuracy as the final clustering accuracy of our algorithm in Figures 5, 6, and Table 1.

To demonstrate the practical applicability of our method relative to mainstream non-SDP approaches, we compare it with three established bi-clustering algorithms: sparseBC Tan & Witten (2014), COBRA Chi et al. (2017), and SSVD Lee et al. (2010). We evaluate performance on both simulated data and a real-world lung cancer gene dataset using bi-clustering accuracy as the primary metric. For the simulated data, the true membership matrices $U^\star$ and $V^\star$ are known, allowing us to compute the Frobenius norm error $\|U - U^\star\|_{\mathrm{F}}$. We use this error to validate Theorem 3 by progressively increasing the sample size $n$ or $m$.

We set the stopping tolerance $\texttt{tol} = 10^{-4}$ and the maximum iteration number $\texttt{maxiter} = 5$ in Algorithm 1. It is observed that the numerical error can actually reach this tolerance within fewer than five iterations in all tested examples. All experiments were conducted on an MacBook Pro laptop (macOS Ventura 13.5.2), with six 2.2 GHz Intel Core i7 CPUs and 16 GB RAM. The codes are publicly available at `https://github.com/Yuxin-LAA-HPC/Alternating-SDP-for-bi-clustering`.

### 5.1 Simulated data

We generate the $n$-by-$m$ data matrices $X = M + N$ with the checkerboard structure. $M$ is a piece-wise constant matrix on $K_R \times K_C$ pieces, as shown in equation 18. $N$ is a noise matrix with $N_{ij} \sim \mathcal{N}(0, \sigma^2)$. We repeat the test ten times by sampling the noise matrix $N$ ten times, and then we report the results based on the mean and confidence interval (error bars) from these ten outputs.

### 5.1.1 Behaviors of the Alternative SDP algorithm

We provide the numerical evidence to support that the error bound in Theorem 3 is tight. To generate $M$, we set $K_R^\star = 2$ with $n_1 = n_2 = 0.5 \cdot n$ and $K_C^\star = 3$ with $m_1 = m_2 = 0.3 \cdot m$, $m_3 = 0.4 \cdot m$. The value of $c_{ij}$ in $M$ is specified as

$$c_{1,1} = 0.2, \quad c_{2,1} = 0.1, \quad c_{1,2} = 0.3, \quad c_{2,2} = 0.4, \quad c_{1,3} = 0.5, \quad c_{2,3} = 0.6. \tag{28}$$

To generate $N$ with $N_{ij} \sim \mathcal{N}(0, \sigma^2)$, we vary the noise level $\sigma$ from $\{0.28, 0.30, 0.32, \ldots, 0.46, 0.48\}$. The size of the data matrices varies as follows:

$$n = 120 + 12i, \quad m = 100 + 10i, \quad i = 1, 2, \ldots, 10.$$

Figure 1 confirms that the error $\|U - U^\star\|_F^2 \sim O(1/n)$, regardless of the different level of noise, where $U$ is the solution computed by Algorithm 1. This is consistent with Theorem 3 and also suggests that the upper bound is tight on the order of $1/n$.

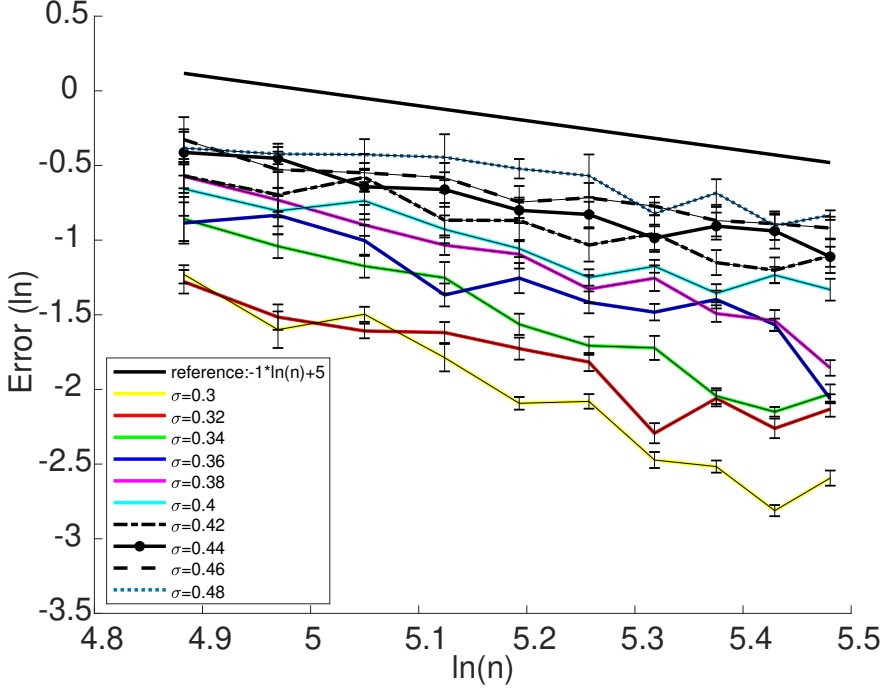

Figure 1: The decay of the error, $\ln\left(\|U - U^\star\|_F^2\right)$, in term of $\ln(n)$, tested on various values of $\sigma$. The "reference" solid black line has the slope $-1$, indicating that the theoretic convergence rate of $\|U - U^\star\|_F$ is $O(1/\sqrt{n})$.

Moreover, we examine how the alternative iteration ($p > 1$) performs better than the one-step method ($p = 1$) when dealing with noisy data. Figure 2 shows the similar error plot to Figure 1, by using the blue lines to represent the iterative scheme and the red lines for the one-step scheme. Since all blue lines are lower than the red lines in the same color, we observe that the effectiveness of iteration for the alternative SDP algorithm.

Next, we demonstrate how various parameters, such as the selection of $(K_R, K_C)$, the dimension $n = m$, and the noise level, affect the convergence of Algorithm 1 in terms of the objective function $F$ in equation 10 and the errors of $U, V$. We generate $M$ by setting $K_R^\star = 4$ with $n_1 = n_2 = 0.2 \cdot n$, $n_3 = n_4 = 0.3 \cdot n$ and $K_C^\star = 3$ with $m_1 = m_2 = 0.3 \cdot m$, $m_3 = 0.4 \cdot m$. The value of $M$ is set as

$$\begin{aligned}
&c_{1,1} = 0.2, \quad c_{2,1} = 0.1, \quad c_{3,1} = 0.3, c_{4,1} = 0.4, \quad c_{1,2} = 0.3, \quad c_{2,2} = 0.4, \\
&c_{3,2} = 0.5, \quad c_{4,2} = 0.2, \quad c_{1,3} = 0.6, c_{2,3} = 0.5, \quad c_{3,3} = 0.4, \quad c_{4,3} = 0.3.
\end{aligned} \tag{29}$$

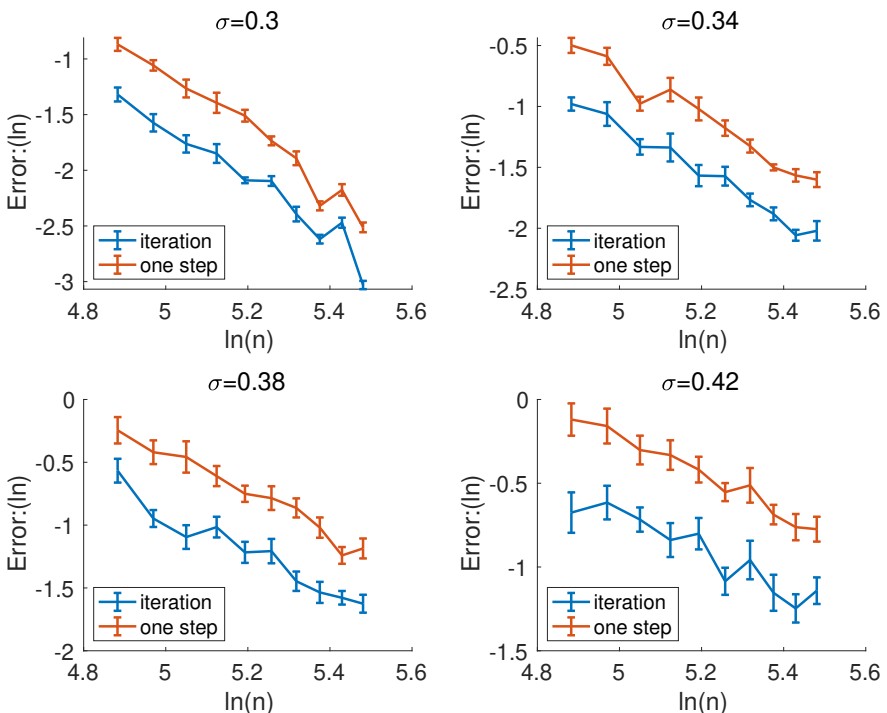

Figure 2: Comparison between the results by the alternating iteration and a single step: $x$-axis and $y$-axis present $\ln(n)$ and the error $\ln\left(\|U - U^\star\|_{\mathsf{F}}^2\right)$, respectively. In these four subplots, we show the results with different noise levels $\sigma = 0.3$, $0.34$, $0.38$, and $0.42$.

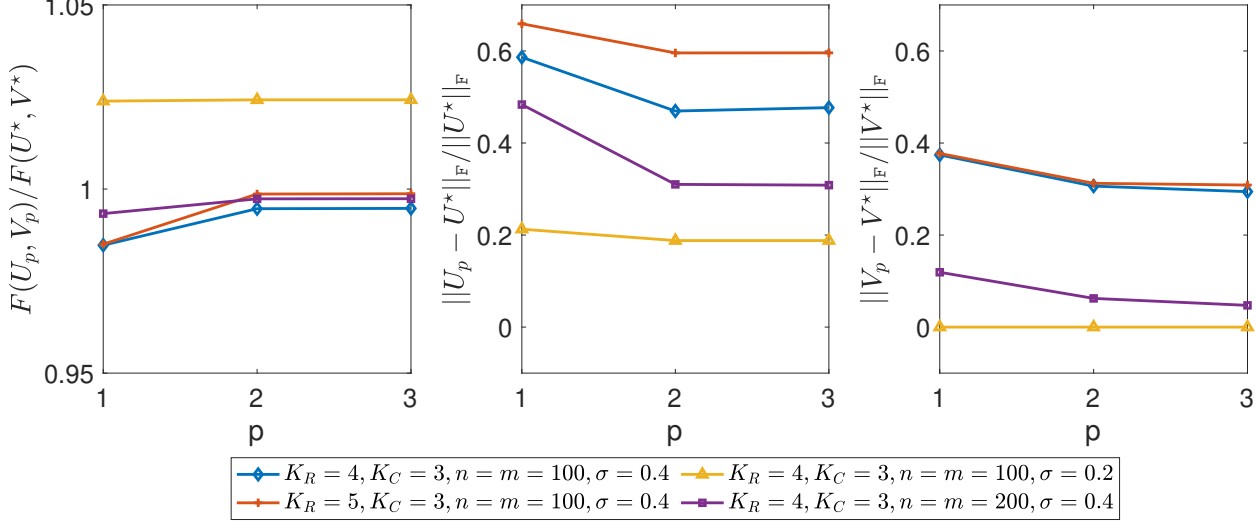

Figure 3: The (rescaled) objective objective function $F_p$ in Algorithm 1 and the errors of $U_p$ and $V_p$ with respect to the iteration $p$: $\frac{F(U_p,V_p)}{F(U^\star,V^\star)}$, $\frac{\|U_p - U^\star\|_{\mathsf{F}}}{\|U^\star\|_{\mathsf{F}}}$, and $\frac{\|V_p - V^\star\|_{\mathsf{F}}}{\|V^\star\|_{\mathsf{F}}}$.

Figure 3 shows the relative value for the objective function $F(U_p, V_p)$ along with the Frobenius norm of $U_p$ and $V_p$. For the lower noise level $\sigma = 0.2$ ( compared with $\sigma = 0.4$) and with the correct choice of $K_R$, we observe that the one-step iteration already yields results very close to those obtained after three steps ($p = 1$ vs. $p = 3$), and the errors in the membership matrices are also very small. When the noise is larger and

$K_R$ is misspecified, additional alternating iterations further improve the minimization of $F$ and reduce the estimation errors.

### 5.1.2 Influence of parameters $K_R$ and $K_C$ on the Alternative SDP algorithm

We illustrate in Figure 4 how different choices of $K_R$ and $K_C$ affect the performance of the Alternative SDP algorithm, i.e., Algorithm 1. The matrix $M$ is generated using $K_R^\star = 2$ row clusters of sizes $n_1 = n_2 = 0.5 \cdot n$ and $K_C^\star = 3$ column clusters with sizes $m_1 = m_2 = 0.3 \cdot m$ and $m_3 = 0.4 \cdot m$, where $n = 240$ and $m = 200$. The noise matrix $N$ is defined entrywise by $N_{ij} \sim \mathcal{N}(0, 0.3^2)$. We test four combinations of $(K_R, K_C)$ no smaller than the true number of rows and columns ($K_R^\star = 2,, K_C^\star = 3$). As shown in Figure 4, the estimated block matrix $\bar{X} = UXV$ is only perturbed by negligible noise, and the checkerboard pattern of $\bar{X}$ clearly remains the same. Indeed, after the rounding step for these different output matrices $\bar{X}$ in Figure 4, we find they all share the same bi-clustering partitions, as well as the clustering accuracies.

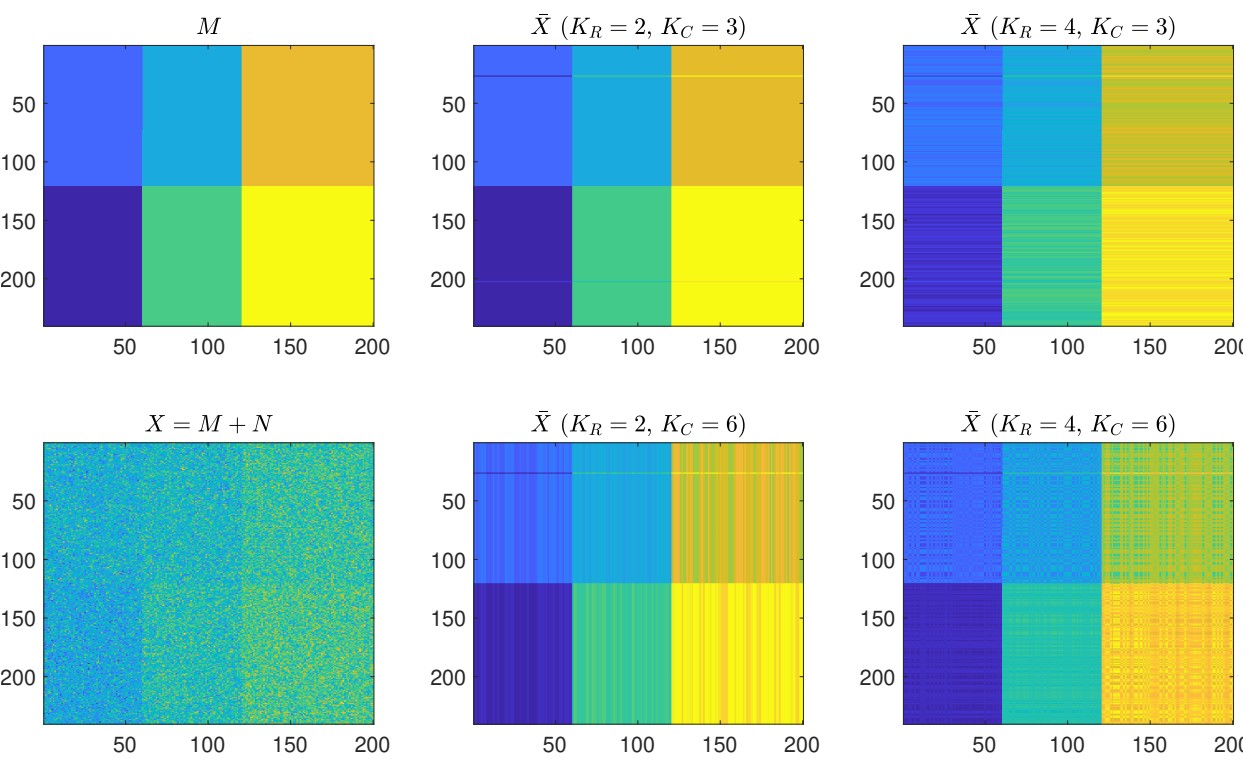

Figure 4: Influence of parameters $K_R$ and $K_C$ on $\bar{X} = UXV$ computed by the Alternative SDP algorithm. In the first column, we display the subfigures corresponding to the matrices $M$ and $X$. In the remaining columns, we present $\bar{X} = UXV$, as computed by the Alternative SDP algorithm with different choices of $K_R$ and $K_C$, i.e., $(K_R, K_C) = (2, 3)$, $(2, 6)$, $(4, 3)$, and $(4, 6)$.

### 5.1.3 Comparison with other algorithms

We compared Algorithm 1, which employs the parameters $K_R = 2$ and $K_C = 3$, with the other three bi-clustering algorithms:

1. sparseBC algorithm: The algorithm proposed by Tan & Witten (2014) is implemented in the `R` package `sparseBC` with the parameters $k = 2$, $r = 3$ and $\lambda$ chosen from $\{0, 1, \ldots, 10\}$.

2. COBRA: This algorithm proposed by Chi et al. (2017), is implemented in the `R` package `cvxbiclustr` to solve a convex bi-clustering problem.

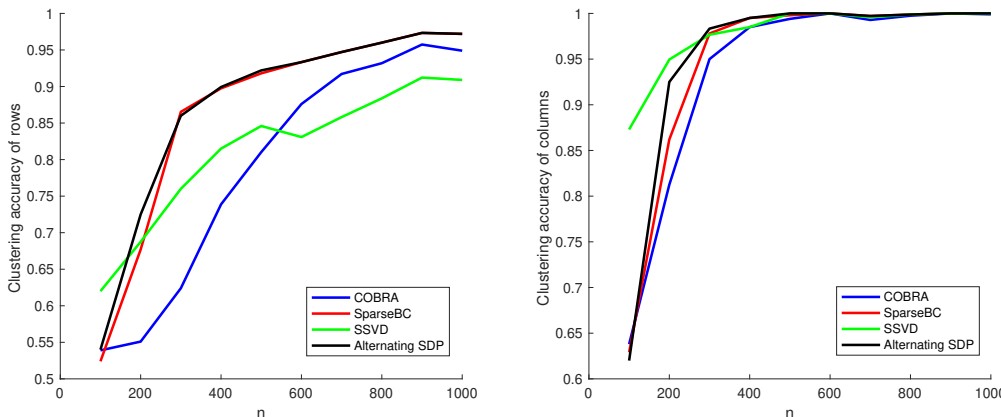

Figure 5: Bi-clustering accuracy by different algorithms with fixed $\sigma = 0.8$: The sub-figures on the left and right show the accuracy of the clustering of the rows and columns, respectively.

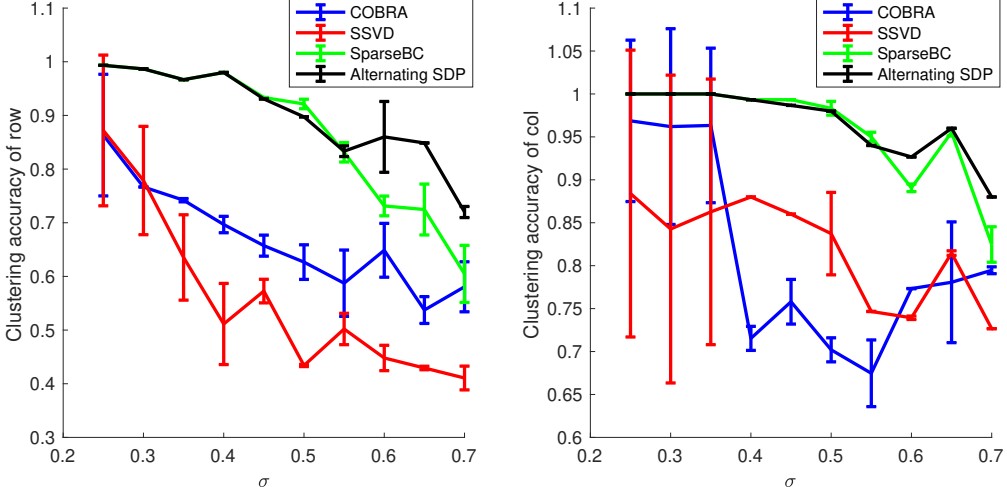

Figure 6: Bi-clustering accuracy by different algorithms with fixed $n = 150$ and $m = 150$: The sub-figures on the left and right show the accuracy of the clustering of the rows and columns, respectively.

3. SSVD: This algorithm is proposed by Lee et al. (2010) based on the singular value decomposition and is implemented in the R package `s4vd` with the parameter about the layer $K = 10$.

**Tests of accuracy**   We first generate $M$ as in equation 28, fix the noise level $\sigma = 0.8$, and change the size of $X$ to $n = m = 100 \cdot i$ for $i = 1, 2, \dots, 10$ in order to compare the different algorithms. As seen in Figure 5, Alternating SDP algorithm outperforms the other three algorithms in terms of clustering accuracy of rows and columns when $n$ is not too small.

We utilize the same $M$ as presented in equation 29 with $K_R^\star = 4$ and $K_C^\star = 3$. The size $n = m = 150$ is fixed and $\sigma$ is set as $\sigma = 0.2 + 0.05 \cdot i$ for $i = 1, 2, \dots, 10$ to compare the clustering accuracy of rows and columns using the different algorithms in Figure 6. From these two figures, our algorithm is shown to be more accurate and stable in most cases.

**Tests of performance**   Next, we compare the performance of Algorithm 1 with three other algorithms implemented in R for large-scale problems. To ensure a fair comparison, we also report the clustering accuracy.

Our implementation is written in `Python`, with the computationally intensive part based on the `Python` code (`https://github.com/simonsfoundation/sdp_kmeans`) of the low-rank SDP solver of Tepper et al. (2018). Note that the publicly available `R` packages of SSVD and COBRA are essentially the wrappers for the `C/C++` numerical routines. To make the comparison across programming environments more consistent, we additionally wrap our `Python` implementation with an `R` interface and conduct all experiments within the `R` environment. As a result, our reported runtime includes the extra overhead from cross-language communication.

The results are summarized in Table 1. They show that the Alternating SDP algorithm is the fastest when $X$ is nearly square, and it still maintains a clear runtime advantage over the native `R` baselines. Thus, the runtime differences in Table 1 largely reflect the computational complexity of the underlying algorithms.

Table 1: The performance and the clustering accuracy of rows and columns by using different algorithms.

| | Size / Algo | $500 \times 500$ | $1000 \times 1000$ | $2000 \times 500$ | $4000 \times 250$ | $2000 \times 2000$ |
|---|---|---|---|---|---|---|
| | Alternating SDP | **4.96** | **9.05** | 28.88 | 82.76 | **26.19** |
| Runtime (sec) | sparseBC | 5.32 | 27.72 | 35.36 | **34.59** | 207.96 |
| | COBRA | 49.25 | 153.17 | 142.77 | 151.48 | 528.13 |
| | SSVD | 7.98 | 13.36 | **23.26** | 56.07 | 32.20 |
| | Alternating SDP | **1** | **1** | **1** | **0.9985** | **1** |
| Accuracy of col | sparseBC | **1** | **1** | **1** | **0.9985** | **1** |
| | COBRA | 1 | 0.681 | 0.6525 | 0.73825 | 1 |
| | SSVD | 0.552 | 0.582 | 0.7655 | 0.6965 | 0.919 |
| | Alternating SDP | **1** | **1** | **1** | **1** | **1** |
| Accuracy of row | sparseBC | **1** | **1** | **1** | **1** | **1** |
| | COBRA | 1 | 1 | 1 | 1 | 1 |
| | SSVD | 0.998 | 1 | 1 | 0.592 | 0.559 |

## 5.2 Application to real-world datasets

### 5.2.1 Application to genomics

For the lung cancer data studied by Busygin et al. (2008), which contains 56 samples and 12,625 genes, we first select subsets of genes with the highest variance—500, 1000, and 3000 genes, respectively—to form the data matrices. Since the true clustering of the 56 samples is known (17 normal subjects, 20 pulmonary carcinoid tumors, 13 colon metastases, and 6 small-cell carcinomas), we have $K_C^\star = 4$. We apply our algorithm using $K_C = K_C^\star = 4$ and $K_R = 10$ and report the sample-clustering accuracy in Table 2. The results show that our method achieves accuracy comparable to that reported in prior studies Tan & Witten (2014); Chi et al. (2017). Additionally, Figure 7 visualizes the column membership matrix $V$ obtained from Algorithm 1, and Figure 8 displays the recovered signal matrix $UXV$ representing gene-expression levels. Both figures further confirm the accuracy and interpretability of our approach.

Table 2: Clustering accuracy on the lung cancer dataset with different number of genes by Alternating SDP algorithm.

| Number of genes | 500 | 1000 | 3000 |
|---|---|---|---|
| Clustering accuracy | 96.4% | 98.2% | 98.2% |

### 5.2.2 Application to gene expression

The Alizadeh-2000-v2 dataset Alizadeh et al. (2000) is a widely used benchmark gene expression dataset in bioinformatics and computational biology, which features 62 samples (patients/cells) and 2094 genes. It includes gene expression measurements from several types of lymphoma samples, including diffuse large B-cell lymphoma (DLBCL), follicular lymphoma (FL), and chronic lymphocytic leukemia (CLL), which means

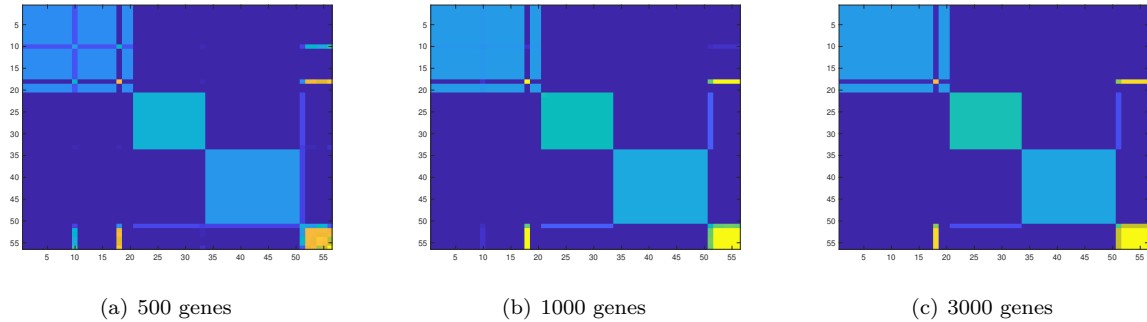

| (a) 500 genes | (b) 1000 genes | (c) 3000 genes |

Figure 7: The column membership matrix $V$ (after applying the proper column permutation for visualization) computed by Alternating SDP algorithm.

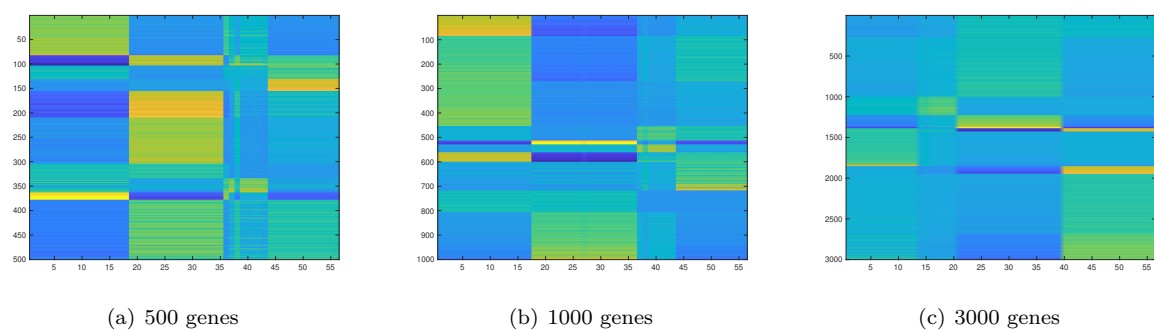

| (a) 500 genes | (b) 1000 genes | (c) 3000 genes |

Figure 8: The predicted $M$ (i.e., $UXV$) with $U$ and $V$ computed by Alternating SDP algorithm.

that the number of column clusters is known to be $K_C^\star = 3$. We apply our algorithm with $K_C = K_C^\star = 3$ and $K_R = 5, 10, 20$, and present the resulting sample-clustering accuracies in Table 3. As shown in Table 3, our algorithm consistently achieves high accuracy (above 96%) for all values of $K_R$.

Table 3: Clustering accuracy on the gene expression dataset with different choices of $K_R$ by Alternating SDP algorithm.

| Row clusters ($K_R$) | 5 | 10 | 20 |
|---|---|---|---|
| Clustering accuracy | 96.8% | 98.4% | 98.4% |

## 6 Conclusion

We propose an alternating SDP algorithm to accurately and efficiently solve the bi-clustering problem. This approach alternates between solving two SDP subproblems, requiring only a mild assumption on the initial values. A key advantage of this iterative formulation is its guaranteed error bound. By leveraging the intrinsic low-rank structure of the problem, our numerical method achieves significant improvements in computational efficiency. Experiments on both simulated and real datasets demonstrate that the accuracy and efficiency of our algorithm are comparable to or exceed those of mainstream methods for the block bi-clustering problem.

## Acknowledgments

The authors thank Xiuyuan Chen for helpful discussions. X. Zhou acknowledges the supported by General Research Funds from the Research Grants Council of the Hong Kong Special Administrative Region, China (Project No. 11318522,11308323, 11304525). W. Gao acknowledges the supported by the National Key R&D Program of China (Grant No. 2021YFA1003305).

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

## A   Proof of Property 1

The proofs of the two conclusions are the same. Thus, we only prove the second result. We start with the KKT conditions of the convex SDP  equation 15 at any optimal solution $\hat{V}$:

$$-A_U + \frac{\lambda \mathbf{1}^\top + \mathbf{1}\lambda^\top}{2} + \mu I - S - Z = 0, \tag{30}$$

$$S \succeq 0, \tag{31}$$

$$Z \geq 0, \tag{32}$$

$$\langle S, \hat{V} \rangle = 0, \tag{33}$$

$$\langle Z, \hat{V} \rangle = 0, \tag{34}$$

where the Lagrange multipliers $A_U$, $S$, $Z$ are $m$-by-$m$ matrices, $\lambda$ is a vector with length $m$, and $\mu$ is a number.

Firstly, we construct a special primal-dual solution which satisfies the above KKT conditions. By $U \succeq 0$, we define $A := \sqrt{U}M$ and write the matrix $A$ in the column form $A = [A_1, A_2, \ldots, A_m]$. Then $A_i = \sqrt{U}M_i$ and $A^\top A = M^\top U M = A_U$.

We shall show that the construction

$$(\lambda^\star, S^\star = 0, Z^\star, \mu^\star = 0, V^\star)$$

where

$$\lambda_i^\star := A_i^\top A_i = M_i^\top U M_i, \quad 1 \leq i \leq m;$$

$$Z_{ij}^\star := \begin{cases} 0, & \exists\, k, \text{ such that } i, j \in E_k, \\ \frac{\|A_i - A_j\|^2}{2} \geq 0, & \text{otherwise,} \end{cases} \quad 1 \leq i, j \leq m \tag{35}$$

and $V^\star$ defined in equation 20, is a primal-dual solution of equation 15. We only need to show the above construction defined by equation 35 satisfies the KKT conditions equation 30–equation 34. Since equation 31–equation 34 are obviously satisfied, we now check equation 30 in the following. By equation 35, we compute the LHS of equation 30 as

$$D := -A_U + \frac{\lambda^\star \mathbf{1}^\top + \mathbf{1}(\lambda^\star)^\top}{2} + \mu^\star I - S^\star - Z^\star = -A^\top A + \frac{\lambda^\star \mathbf{1}^\top + \mathbf{1}(\lambda^\star)^\top}{2} - Z^\star.$$

Equivalently,

$$D_{ij} = A_i^\top A_j + \frac{A_i^\top A_i + A_j^\top A_j}{2} - Z_{ij}^\star = M_i^\top U M_j + \frac{M_i^\top U M_i + M_j^\top U M_j}{2} - Z_{ij}^\star. \tag{36}$$

Consider any index pair $(i, j)$. If they are in the same column cluster $k$, i.e., $i, j \in E_k$, then $M_i = M_j$ and it follows that $D_{ij} = 0$ by the definition of $Z^\star$ in equation 35. Otherwise, by the definition of $Z^\star$ in equation 35 again,

$$D_{ij} = -A_i^\top A_j + \frac{A_i^\top A_i + A_j^\top A_j}{2} - \frac{\|A_i - A_j\|^2}{2} = 0.$$

Therefore, we have verified that $(\lambda^\star, S^\star, Z^\star, \mu^\star, V^\star)$ also satisfies equation 30. Therefore, $V^\star$ is an optimal solution of the Problem equation 15.

Secondly, we show $V^\star$ is the unique solution of equation 15 for any $U$ satisfying equation 24. The Lagrangian function for the sub-problem equation 15 (by considering $\min \langle -A_U, V \rangle$ ) is given by

$$L(V, \lambda, \mu, S, Z) = \langle -A_U, V \rangle + \lambda^\top (V\mathbf{1} - \mathbf{1}) + \mu(\langle I, V \rangle - K_C) - \langle S + Z, V \rangle.$$

Note that for any $\lambda, \mu, S, Z$ and a feasible $V \in S_V$, the Lagrangian $L(V, \lambda, \mu, S, Z) = -\langle A_U + S + Z, V \rangle$ by using the equality constraints in $S_V$. In particular, for the optimal solution $(V^\star, \lambda^\star, \mu^\star, S^\star, Z^\star)$ we have constructed to satisfy the KKT conditions equation 31 and equation 34, we have $L(V^\star, \lambda^\star, \mu^\star, S^\star, Z^\star) = -\langle A_U, V^\star \rangle$.

If there is another optimal solution $\hat{V}$ of equation 15, then $\langle -A_U, \hat{V} \rangle \le \langle -A_U, V^\star \rangle$. We shall show that this $\hat{V}$ has to be the same as $V^\star$. Since the KKT pair $(V^\star, \lambda^\star, \mu^\star, S^\star, Z^\star)$ solves the min-max problem for the Lagrangian function, we have that

$$\begin{aligned} -\langle A_U, V^\star \rangle = L(V^\star, \lambda^\star, \mu^\star, S^\star, Z^\star) &\le L(\hat{V}, \lambda^\star, \mu^\star, S^\star, Z^\star) = -\langle A_U + S^\star + Z^\star, \hat{V} \rangle \\ &\le -\langle A_U, \hat{V} \rangle \le -\langle A_U, V^\star \rangle, \end{aligned} \tag{37}$$

where we use $\langle S^\star + Z^\star, \hat{V} \rangle \ge 0$ due to equation 31, equation 34 and $\hat{V} \ge 0$, $\hat{V} \succeq 0$.

Equality equation 37 in fact implies that $\langle S^\star + Z^\star, \hat{V} \rangle = 0$. From the fact that $S^\star = 0$, $Z^\star \ge 0$ and $\hat{V} \succeq 0$, we have

$$\langle Z^\star, \hat{V} \rangle = \sum_{ij} Z_{ij}^\star \hat{V}_{ij} = 0. \tag{38}$$

Based on this and the condition equation 24, we show that $\hat{V} = V^\star$ as follows.

Note that the assumption equation 24 can guarantee $Z_{ij}^\star = \|A_i - A_j\|^2/2 > 0$ when $i \in E_k$, $j \in E_l$ with $k \ne l$. Thus, equation 38 gives $\hat{V}_{ij} = 0$, if $i \in E_k$, $j \in E_l$ with $k \ne l$, and consequently $\hat{V}$ is a block diagonal matrix of the following form:

$$\hat{V} = \begin{bmatrix} \hat{V}_1 & 0 & \cdots & 0 \\ 0 & \hat{V}_2 & \cdots & 0 \\ \vdots & \vdots & \ddots & \vdots \\ 0 & 0 & \cdots & \hat{V}_{K_C} \end{bmatrix}.$$

Together with $\hat{V}\mathbf{1} = \mathbf{1}$, we have $\hat{V}_i \mathbf{1} = \mathbf{1}$ for any $i = 1, 2, \ldots, K_C$. It means that $1$ must be an eigenvalue of $\hat{V}_i$ and $\mathrm{Tr}(\hat{V}_i) \ge 1$. Then by $\mathrm{Tr}(\hat{V}) = \sum_{i=1}^{K_C} \mathrm{Tr}(\hat{V}_i) = K_C$, we obtain that the other eigenvalues of $\hat{V}_i$ should be $0$. Thus, each block $\hat{V}_i$ has rank one. Therefore, $\hat{V}$ has to take the same form as $V^\star$, which means that $V^\star$ is the unique solution of equation 15.

## B  Proof of Theorem 2

To prove Theorem 2, we first show the following lemma to estimate the bound of $\|U^\star - U\|_{\mathsf{F}}^2$.

**Lemma 1.** *Write the matrix $M$ in row-wise form, $M^\top = [(M^\top)_1, \ldots, (M^\top)_n]$. Then*

$$\|U^\star - U\|_{\mathsf{F}}^2 \le \frac{4}{n_{\min} \cdot \text{gap}_r^2(M)} \big(\text{Tr}(MV^\star M^\top U^\star) - \text{Tr}(MVM^\top U)\big) \tag{39}$$

*holds for all $U \in S_U$ and $V \in S_V$, where $S_U$ and $S_V$ are defined in equation 6 and equation 7, and*

$$\text{gap}_r(M) := \min_{i \in C_k, j \in C_l, k \ne l} \|(M^\top)_i - (M^\top)_j\|_2. \tag{40}$$

**Remark 6.** *We have a similar result for the bound of $\|V - V^\star\|_{\mathsf{F}}^2$. Lemma 1 also holds for all $U \in D_U$ and $V \in D_V$, since $D_U \subseteq S_U$ and $D_V \subseteq S_V$.*

*Proof.* Note that $U$ and $U^\star$ are positive semidefinite and stochastic matrix whose columns are probability vectors. It follows by Perron–Frobenius theorem that their eigenvalues all line in $[0, 1]$. Therefore,

$$\|U\|_{\mathsf{F}}^2 = \sum_j \lambda_j^2 \le \sum_j \lambda_j = \text{Tr}(U). \tag{41}$$

This fact together with $(U^\star)^2 = U^\star$ implies that

$$\|U^\star - U\|_{\mathsf{F}}^2 = \|U^\star\|_{\mathsf{F}}^2 + \|U\|_{\mathsf{F}}^2 - 2\text{Tr}(UU^\star) \le 2K_R - 2\text{Tr}(UU^\star) = 2\text{Tr}((U^\star - U)U^\star). \tag{42}$$

Let $\Omega$ denote the indices of a $n \times m$ matrix for the diagonal blocks specified the ground truth, and $\Omega^c$ the indices in the off-diagonal blocks, and $\Omega_t$ denote the indices in the diagonal block for the cluster $t$. So $\Omega = \cup_t \Omega_t$. For example, we write $\mathbf{1}\mathbf{1}^\top$ as the sum of two matrices $(\mathbf{1}\mathbf{1}^\top)_\Omega$ and $(\mathbf{1}\mathbf{1}^\top)_{\Omega^c}$, defined by

$$(\mathbf{1}\mathbf{1}^\top)_\Omega := \begin{bmatrix} \mathbf{1}_{n_1}\mathbf{1}_{n_1}^\top & 0 & \cdots & 0 \\ 0 & \mathbf{1}_{n_2}\mathbf{1}_{n_2}^\top & \cdots & 0 \\ \vdots & \vdots & \ddots & \vdots \\ 0 & 0 & \cdots & \mathbf{1}_{n_{K_R}}\mathbf{1}_{n_{K_R}}^\top \end{bmatrix}$$

and

$$(\mathbf{1}\mathbf{1}^\top)_{\Omega^c} := \begin{bmatrix} 0 & \mathbf{1}_{n_1}\mathbf{1}_{n_2}^\top & \cdots & \mathbf{1}_{n_1}\mathbf{1}_{n_{K_R}}^\top \\ \mathbf{1}_{n_2}\mathbf{1}_{n_1}^\top & 0 & \cdots & \mathbf{1}_{n_2}\mathbf{1}_{n_{K_R}}^\top \\ \vdots & \vdots & \ddots & \vdots \\ \mathbf{1}_{n_1}\mathbf{1}_{n_{K_R}}^\top & \mathbf{1}_{n_2}\mathbf{1}_{n_{K_R}}^\top & \cdots & 0 \end{bmatrix}.$$

In addition, $(\mathbf{1}\mathbf{1}^\top)_{\Omega_t}$ refers to the matrix in which all elements are zero, apart from the $t$-th diagonal block elements which take the value of 1 at each entry. With this notation, the matrix $U^\star$ is then rewritten as

$$n_{\min}U^\star = \mathbf{1}\mathbf{1}^\top - (\mathbf{1}\mathbf{1}^\top)_{\Omega^c} - \sum_{t=1}^{K_R}(1 - \frac{n_{\min}}{n_t})(\mathbf{1}\mathbf{1}^\top)_{\Omega_t}. \tag{43}$$

Note that $(U^\star - U)(\mathbf{1}\mathbf{1}^\top) = 0$ implied by $(U^\star - U)\mathbf{1} = \mathbf{1} - \mathbf{1} = 0$. This further means that

$$0 = (U^\star - U)_\Omega(\mathbf{1}\mathbf{1}^\top) + (U^\star - U)_{\Omega_c}(\mathbf{1}\mathbf{1}^\top) = (U^\star - U)_\Omega(\mathbf{1}\mathbf{1}^\top) - U_{\Omega_c}(\mathbf{1}\mathbf{1}^\top).$$

and thus

$$(U^\star - U)_\Omega(\mathbf{1}\mathbf{1}^\top) = U_{\Omega_c}(\mathbf{1}\mathbf{1}^\top) \ge 0,$$

which implies for each block

$$(U^\star - U)_{\Omega_t}(\mathbf{1}\mathbf{1}^\top) \ge 0, \quad \forall t.$$

From this inequality and equation 43, we have the following bounds for equation 42

$$
\begin{aligned}
\|U^\star - U\|_{\mathsf{F}}^2 &\leq 2\operatorname{Tr}((U^\star - U)U^\star) \\
&= -\frac{2}{n_{\min}}\operatorname{Tr}\left((U^\star - U)\big((\mathbf{1}\mathbf{1}^\top)_{\Omega^c} + \sum_{t=1}^{K_R}(1 - \frac{n_{\min}}{n_t})(\mathbf{1}\mathbf{1}^\top)_{\Omega_t}\big)\right) \\
&\leq -\frac{2}{n_{\min}}\operatorname{Tr}((U^\star - U)(\mathbf{1}\mathbf{1}^\top)_{\Omega^c}).
\end{aligned}
\tag{44}
$$

Let $R := MM^\top$ and write $R = D + C$, where

$$
D := \frac{1}{2}r\mathbf{1}^\top + \frac{1}{2}\mathbf{1}r^\top \quad \text{and} \quad C := R - D
$$

with the column vector $r := \operatorname{diag}(R)$. Note that $C_{ij} = R_{ij} - \frac{1}{2}(R_{ii} + R_{jj}) = -\frac{1}{2}\|(M^\top)_i - (M^\top)_j\|^2 \leq 0$, and $C$ satisfies $C_\Omega = 0$ and $C_{\Omega^c} = C$. Let $|C|_{\min}$ denote the minimal absolute value among elements of the matrix $C$, then $(\mathbf{1}\mathbf{1}^\top)_{\Omega^c} \leq -\frac{C}{|C|_{\min}}$. By this inequality and the fact that $-(U^\star - U)_{\Omega^c} = U_{\Omega^c} \geq 0$, we have that

$$
\begin{aligned}
-\operatorname{Tr}((U^\star - U)_{\Omega^c}(\mathbf{1}\mathbf{1}^\top)_{\Omega^c}) &\leq \frac{1}{|C|_{\min}}\operatorname{Tr}((U^\star - U)_{\Omega^c}C) \\
&= \frac{1}{|C|_{\min}}\operatorname{Tr}((U^\star - U)C_{\Omega^c}) \\
&= \frac{1}{|C|_{\min}}\operatorname{Tr}((U^\star - U)C),
\end{aligned}
\tag{45}
$$

where the last two equalities are due to the simple fact $\operatorname{Tr}(AB_{\Omega^c}) = \operatorname{Tr}(A_{\Omega^c}B) = \operatorname{Tr}(A_{\Omega^c}B_{\Omega^c})$ for any two matrices in $\mathbb{R}^{n\times n}$.

Then by following equation 44, we have that

$$
\begin{aligned}
\|U^\star - U\|_{\mathsf{F}}^2 &\leq -\frac{2}{n_{\min}}\operatorname{Tr}((U^\star - U)(\mathbf{1}\mathbf{1}^\top)_{\Omega^c}) \\
&\leq \frac{2}{n_{\min}|C|_{\min}}\operatorname{Tr}((U^\star - U)C) \\
&= \frac{2}{n_{\min}|C|_{\min}}\operatorname{Tr}((U^\star - U)R),
\end{aligned}
\tag{46}
$$

where the last step uses $R = C + D$ and the following property about the matrix $D$:

$$
2\operatorname{Tr}\big((U^\star - U)D\big) = \operatorname{Tr}\big((U^\star - U)(r\mathbf{1}^\top + \mathbf{1}r^\top)\big) = 2\operatorname{Tr}\big((U^\star - U)\mathbf{1}r^\top\big) = 0,
$$

since $(U^\star - U)\mathbf{1} = 0$.

Lastly, for the right hand side in equation 46, we note that $\operatorname{Tr}(UR) = \operatorname{Tr}(MM^\top U) = \operatorname{Tr}(MV^\star M^\top U)$. Therefore, by Property 1 about the optimality of $V^\star$ for the sub-problem equation 15 (with $X = M$), it holds that

$$
\begin{aligned}
\operatorname{Tr}((U^\star - U)R) &= \operatorname{Tr}(MV^\star M^\top U^\star) - \operatorname{Tr}(MV^\star M^\top U) \\
&\leq \operatorname{Tr}(MV^\star M^\top U^\star) - \operatorname{Tr}(MVM^\top U)
\end{aligned}
\tag{47}
$$

for any $U \in S_U$ and $V \in S_V$, which completes the proof in view of equation 46 and that $|C|_{\min} = \operatorname{gap}_r(M)/2$. $\qquad\square$

Now we have prepared to prove Theorem 2.

*Proof of Theorem 2.* From Lemma 1, we have

$$\|U^\star - U_1\|_{\mathsf{F}}^2 \leq \frac{4}{n_{\min} \cdot \mathrm{gap}_r^2(M)} \big(\mathrm{Tr}(MV^\star M^\top U^\star) - \mathrm{Tr}(MIM^\top U_1)\big). \tag{48}$$

The proof is to apply Lemma 1 by first estimating the bound $\mathrm{Tr}(MV^\star M^\top U^\star) - \mathrm{Tr}(MIM^\top U_1)$.

Note that $U_1$ is the optimal solution of Problem equation 14 or equation 16 corresponding to the initial $V_0 = I$, we have $F(V_0, U^\star) \leq F(V_0, U_1)$, i.e., $\mathrm{Tr}(XIX^\top U^\star) \leq \mathrm{Tr}(XIX^\top U_1)$, which leads to, by using $X = M + N$,

$$\begin{aligned}
\mathrm{Tr}(MM^\top U^\star) - \mathrm{Tr}(MM^\top U_1) \leq{} &- \mathrm{Tr}(NM^\top U^\star) + \mathrm{Tr}(NM^\top U_1) - \mathrm{Tr}(NN^\top U^\star) \\
&+ \mathrm{Tr}(NN^\top U_1) - \mathrm{Tr}(MN^\top U^\star) + \mathrm{Tr}(MN^\top U_1).
\end{aligned} \tag{49}$$

Together with the fact $MV^\star = M$, the above inequality gives that

$$\begin{aligned}
\mathrm{Tr}(MV^\star M^\top U^\star) - \mathrm{Tr}(MIM^\top U_1) &= \mathrm{Tr}(MIM^\top U^\star) - \mathrm{Tr}(MIM^\top U_1) \\
&\leq |\mathrm{Tr}(NM^\top(U_1 - U^\star)| + |\mathrm{Tr}(MN^\top(U_1 - U^\star)| \\
&\quad + |\mathrm{Tr}(NN^\top(U_1 - U^\star))| \\
&\leq 2\|N\|_2\|M\|_{\mathsf{F}}\|U_1 - U^\star\|_{\mathsf{F}} + \|N\|_2^2\|I\|_{\mathsf{F}}\|U_1 - U^\star\|_{\mathsf{F}} \\
&= \|N\|_2\big(2\|M\|_{\mathsf{F}} + \sqrt{m}\|N\|_2\big)\|U_1 - U^\star\|_{\mathsf{F}}.
\end{aligned} \tag{50}$$

The second inequality above is due to the matrix-norm inequality

$$\mathrm{Tr}(ABC) \leq \|AB\|_{\mathsf{F}}\|C\|_{\mathsf{F}} \leq \|A\|_2\|B\|_{\mathsf{F}}\|C\|_{\mathsf{F}}. \tag{51}$$

Therefore, by equation 48 and equation 50, we have that

$$\|U^\star - U_1\|_{\mathsf{F}} \leq \frac{4\|N\|_2\big(2\|M\|_{\mathsf{F}} + \sqrt{m}\|N\|_2\big)}{n_{\min} \cdot \mathrm{gap}_r^2(M)}. \tag{52}$$

It remains to show the bounds for $\|N\|_2$, $\|M\|_{\mathsf{F}}$, and $\mathrm{gap}_r^2(M)$.

By the random matrix theory (Vershynin, 2012, Corollary 5.35) for the $n \times m$ Gaussian random matrix, we have that with probability $\geq 1 - \eta$,

$$\|N\|_2 \leq \big(\sqrt{n} + \sqrt{m} + \sqrt{2\ln\frac{2}{\eta}}\big)\sigma. \tag{53}$$

In addition,

$$\|M\|_{\mathsf{F}} = \sqrt{\sum_{i=1,\dots,n, j=1,\dots,m} M_{ij}^2} \leq \sqrt{mn} \max_{k=1,\dots,K_R; l=1,\dots,K_C} |c_{k,l}|, \tag{54}$$

and

$$\begin{aligned}
\mathrm{gap}_r^2(M) &= \min_{i\in C_k, j\in C_l, k\neq l} \|(M^\top)_i - (M^\top)_j\|_2^2 \\
&= \min_{i\in C_k, j\in C_l, k\neq l} \sum_{t=1}^{K_C} m_t(M_{it} - M_{jt})^2 \\
&\geq \min_{i\in C_k, j\in C_l, k\neq l} \big(m_{\min} \max_{t=1,\dots,K_C}(M_{it} - M_{jt})^2\big) \\
&= m_{\min}\Delta_r^2.
\end{aligned} \tag{55}$$

The conclusion is then proven by combining equation 52, equation 53, equation 54, and equation 55. $\qquad\square$

## C  Proof of Theorem 3

To prove Theorem 3, we establish the following lemma.

**Lemma 2.** *For any $\eta > 0$, with the probability $\geq 1 - \eta$, it holds that*

$$F(U^\star, V^\star) - F(U_p, V_{p-1}) \leq \alpha \delta_r \sigma^2 n, \quad p \geq 2. \tag{56}$$

*Therefore,*

$$F(U^\star, V^\star) - F(U_p, V_p) \leq F(U^\star, V^\star) - F(U_p, V_{p-1}) \leq \alpha \delta_r \sigma^2 n \tag{57}$$

*with*

$$\alpha := \frac{\alpha_0 (\sqrt{m} + \sqrt{K_C})(\sqrt{n} + \sqrt{m} + \sqrt{2 \ln \frac{2}{\eta}})^2}{n\sqrt{n}}.$$

*If $m/m_{\min} = \Theta(1)$, $n/n_{\min} = \Theta(1)$, $n/m = \Theta(1)$ and $\ln(2/\eta) = O(\sqrt{mn})$, as in Remark 2, then $\alpha_0 = \Theta(1)$ and $\alpha = \Theta(1)$.*

*Proof.* Note that $U_1$ is the optimal solution of Problem equation 14 or equation 16 corresponding to the initial $V_0 = I$. Thus, we have $F(V_0, U^\star) \leq F(V_0, U_1)$, i.e.,

$$\mathrm{Tr}(XIX^\top U^\star) \leq \mathrm{Tr}(XIX^\top U_1). \tag{58}$$

Similarly, $V_1$ satisfies $F(V^\star, U_1) \leq F(V_1, U_1)$, i.e.,

$$\mathrm{Tr}(XV^\star X^\top U_1) \leq \mathrm{Tr}(XV_1 X^\top U_1). \tag{59}$$

By $X = M + N$ and equation 5, it is easy to verify that for any $n$-by-$n$ matrix $U$,

$$\mathrm{Tr}(XV^\star X^\top U) = \mathrm{Tr}(XIX^\top U) + \mathrm{Tr}(NV^\star N^\top U) - \mathrm{Tr}(NIN^\top U). \tag{60}$$

Then it follows that

$$
\begin{aligned}
& F(U^\star, V^\star) - F(U_p, V_{p-1}) \\
&= \mathrm{Tr}(XV^\star X^\top U^\star) - \mathrm{Tr}(XV_{p-1} X^\top U_p) \\
&\overset{equation\ 60}{=} \mathrm{Tr}(XIX^\top U^\star) - \mathrm{Tr}(XV_{p-1} X^\top U_p) + \mathrm{Tr}(NV^\star N^\top U^\star) - \mathrm{Tr}(NIN^\top U^\star) \\
&\overset{equation\ 58}{\leq} \mathrm{Tr}(XIX^\top U_1) - \mathrm{Tr}(XV_{p-1} X^\top U_p) + \mathrm{Tr}(NV^\star N^\top U^\star) - \mathrm{Tr}(NIN^\top U^\star) \\
&\overset{equation\ 60}{=} \mathrm{Tr}(XV^\star X^\top U_1) - \mathrm{Tr}(XV_{p-1} X^\top U_p) + \mathrm{Tr}(NV^\star N^\top (U^\star - U_1)) \\
&\quad - \mathrm{Tr}(NIN^\top (U^\star - U_1)) \\
&\overset{equation\ 59}{\leq} \mathrm{Tr}(XV_1 X^\top U_1) - \mathrm{Tr}(XV_{p-1} X^\top U_p) + \|N\|_2^2 (\sqrt{K_C} + \sqrt{m}) \|U^\star - U_1\|_{\mathsf{F}} \\
&\overset{equation\ 25}{\leq} \|N\|_2^2 (\sqrt{K_C} + \sqrt{m}) \|U^\star - U_1\|_{\mathsf{F}}.
\end{aligned}
\tag{61}
$$

The second last inequality uses the matrix norm inequality equation 51, as well as $\|V^\star\|_{\mathsf{F}} = \sqrt{K_C}$ and $\|I\|_{\mathsf{F}} = \sqrt{m}$.

With the bound of $\|N\|_2$ in equation 53 and the bound of $\|U^\star - U_1\|_{\mathsf{F}}$ in Theorem 2, equation 61 completes the proof. $\qquad\square$

*Proof of Theorem 3.* To bound the error $E_p^{(U)} := \|U^\star - U_p\|_{\mathsf{F}}$, we use Lemma 1 to have

$$\left(E_p^{(U)}\right)^2 \leq \frac{4}{n_{\min} \cdot \mathrm{gap}_r^2(M)} \left(\mathrm{Tr}(MV^\star M^\top U^\star) - \mathrm{Tr}(MV_{p-1} M^\top U_p)\right). \tag{62}$$

Then we estimate $\text{Tr}(MV^\star M^\top U^\star) - \text{Tr}(MV_{p-1}M^\top U_p)$. Since $U_p$ is an optimal solution of Problem equation 14 or equation 16 when fixing $V = V_{p-1}$, by using $M = X - N$, we obtain

$$\text{Tr}(MV^\star M^\top U^\star) - \text{Tr}(MV_{p-1}M^\top U_p) \le e_1 + e_2 + e_3 + F(U^\star, V^\star) - F(U_p, V_{p-1}), \tag{63}$$

where

$$e_1 := |\text{Tr}(NV^\star M^\top U^\star) - \text{Tr}(NV_{p-1}M^\top U_p)|,$$
$$e_2 := |\text{Tr}(MV^\star N^\top U^\star) - \text{Tr}(MV_{p-1}N^\top U_p)|,$$
$$e_3 := |\text{Tr}(NV^\star N^\top U^\star) - \text{Tr}(NV_{p-1}N^\top U_p)|.$$

The bounds for the three terms $e_1$, $e_2$, $e_3$ in equation 63 are analyzed below. By using $V^\star M^\top = M^\top$ and equation 51, we have that

$$e_1 \le |\text{Tr}(NV^\star M^\top U^\star) - \text{Tr}(NV^\star M^\top U_p)| + |\text{Tr}(NV^\star M^\top U_p) - \text{Tr}(NV_{p-1}M^\top U_p)|$$
$$\le \|N\|_2\Big(\|M\|_\mathsf{F}\|U^\star - U_p\|_\mathsf{F} + \|U_p M\|_\mathsf{F}\|V^\star - V_{p-1}\|_\mathsf{F}\Big)$$
$$\le \|N\|_2\|M\|_\mathsf{F}\Big(E_p^{(U)} + \sqrt{K_R}\|V^\star - V_{p-1}\|_\mathsf{F}\Big).$$

The last inequality is obtained by $\|U_p\|_\mathsf{F}^2 \le \text{Tr}(U_p) = K_R$ from equation 41. Similarly,

$$e_2 \le \|N\|_2\|M\|_\mathsf{F}\big(E_p^{(U)} + \sqrt{K_R}\|V^\star - V_{p-1}\|_\mathsf{F}\big)$$

and

$$e_3 \le \|N\|_2^2\big(\sqrt{K_C}E_p^{(U)} + \sqrt{K_R}\|V^\star - V_{p-1}\|_\mathsf{F}\big).$$

Therefore, equation 62 leads to the quadratic inequality:

$$(E_p^{(U)})^2 - aE_p^{(U)} - (bE_{p-1}^{(V)} + c_p) \le 0 \tag{64}$$

and similarly

$$(E_{p-1}^{(V)})^2 - a'E_{p-1}^{(V)} - (b'E_{p-1}^{(U)} + c'_{p-1}) \le 0, \tag{65}$$

where $a$, $b$, $c_p$, $a'$, $b'$, and $c'_{p-1}$ are defined as following:

$$
\begin{aligned}
a &:= \frac{4\sqrt{n}\|N\|_2\big(2\|M\|_\mathsf{F} + \sqrt{K_C}\|N\|_2\big)}{n_{\min}m_{\min}\Delta_r^2} \le \frac{2\alpha_a}{\sqrt{n}}\delta_r,\\
b &:= \frac{4\sqrt{K_R}}{n_{\min}m_{\min}\Delta_r^2}\|N\|_2\big(2\|M\|_\mathsf{F} + \|N\|_2\big) \le \frac{\alpha_b}{\sqrt{n}}\delta_r,\\
c_p &:= \frac{4}{n_{\min}m_{\min}\Delta_r^2}\big(F(U^\star, V^\star) - F(U_p, V_{p-1})\big),\\
a' &:= \frac{4\|N\|_2\big(2\|M\|_\mathsf{F} + \sqrt{K_R}\|N\|_2\big)}{n_{\min}m_{\min}\Delta_c^2} \le \frac{2\beta_a}{\sqrt{n}}\delta_r,\\
b' &:= \frac{4\sqrt{K_C}}{n_{\min}m_{\min}\Delta_c^2}\|N\|_2\big(2\|M\|_\mathsf{F} + \|N\|_2\big) \le \frac{\beta_b}{\sqrt{n}}\delta_r,\\
c'_{p-1} &:= \frac{4}{n_{\min}m_{\min}\Delta_c^2}\big(F(U^\star, V^\star) - F(U_{p-1}, V_{p-1})\big).
\end{aligned}
\tag{66}
$$

Here $c_p$ and $c'_p$ decrease with $p$. Furthermore, by Lemma 2, $c_p$ and $c'_p$ can be bounded by

$$c_p \le \tilde{c} := \frac{4}{n_{\min}m_{\min}\Delta_r^2}\alpha\delta_r\sigma^2 n = \frac{\alpha_c^2}{n}\delta_r^2, \tag{67}$$

$$c'_{p-1} \le \tilde{c}' := \frac{4}{n_{\min}m_{\min}\Delta_c^2}\alpha\delta_r\sigma^2 n = \frac{\beta_c^2}{n}\delta_r^2. \tag{68}$$

From the quadratic inequality equation 64, $E_p^{(U)}$ can be bounded by

$$E_p^{(U)} \leq \frac{a + \sqrt{a^2 + 4(bE_{p-1}^{(V)} + c_p)}}{2}. \tag{69}$$

This means that we only need to bound $E_{p-1}^{(V)}$, which can be bounded by $E_{p-1}^{(U)}$ from equation 68, i.e.,

$$E_{p-1}^{(V)} \leq \frac{a' + \sqrt{a'^2 + 4(b'E_{p-1}^{(U)} + \tilde{c}')}}{2} \leq a' + \sqrt{b'E_{p-1}^{(U)}} + \sqrt{\tilde{c}'}. \tag{70}$$

Thus, the problem transfers to bound $E_{p-1}^{(U)}$. From equation 67 and equation 68, we similarly have

$$E_{p-1}^{(U)} \leq a + \sqrt{bE_{p-2}^{(V)}} + \sqrt{\tilde{c}},$$
$$E_{p-2}^{(V)} \leq a' + \sqrt{b'E_{p-2}^{(U)}} + \sqrt{\tilde{c}'}. \tag{71}$$

Furthermore, we obtain the relation between $E_{p-1}^{(U)}$ and $E_{p-2}^{(U)}$:

$$E_{p-1}^{(U)} \leq a + \sqrt{ba' + b\sqrt{b'E_{p-2}^{(U)}} + b\sqrt{\tilde{c}'}} + \sqrt{\tilde{c}} \leq a + b^{\frac{1}{2}}a'^{\frac{1}{2}} + b^{\frac{1}{2}}\tilde{c}'^{\frac{1}{4}} + \tilde{c}^{\frac{1}{2}} + b^{\frac{1}{2}}b'^{\frac{1}{4}}(E_{p-2}^{(U)})^{\frac{1}{4}}. \tag{72}$$

Then we have the bound of $E_{p-1}^{(U)}$:

$$E_{p-1}^{(U)} \leq \frac{\gamma_U}{\sqrt{n}}\delta_r \tag{73}$$

with $\gamma_U = 5\max(\alpha_0, \alpha_a, \alpha_b, \alpha_c, \beta_a, \beta_b, \beta_c)$, by induction. Theorem 2 implies that equation 73 holds for $p = 2$. Then we aim to prove equation 73 holds for $p = i + 1$ if assuming equation 73 holds for $p = i$. Substituting $a, b, \tilde{c}, a', b'$, and $\tilde{c}'$ in equation 72 with equation 66, equation 67, and equation 68, it follows that

$$E_{p-1}^{(U)} \leq \frac{\delta_r}{\sqrt{n}}\left(\alpha_a + \sqrt{\alpha_b\beta_a} + \sqrt{\alpha_b\beta_c} + \alpha_c + \sqrt{\alpha_b\sqrt{\beta_b\gamma_U}}\right) \leq \frac{\delta_r}{\sqrt{n}}\gamma_U,$$

which gives the of $E_p^{(U)}$ together with equation 70 and equation 69. Following a similar way to the bound $\|U^\star - U_p\|_{\mathsf{F}}$, we can also bound $\|V^\star - V_p\|_{\mathsf{F}}$.

$\square$

## D    ADMM for solving (14) and (15)

We present the ADMM algorithm proposed by Wen et al. (2010) to solve equation 14. The solver for equation 15 is similar. The dual of equation 14 is

$$\min_{y \in \mathcal{R}^{n+1}, S \in \mathcal{S}^n, Z \in \mathcal{S}^n} -b^\top y, \tag{74}$$
$$\text{s.t.} \quad \mathcal{A}^*(y) + S + Z = -A_V, S \succeq 0, \quad Z \geq 0.$$

where $b = [\mathbf{1}, K_R]$, $\mathcal{A}^*(y) = \frac{\mathbf{1}y_{1:n}^\top + y_{1:n}\mathbf{1}^\top}{2} + y_{n+1}I_n$, and $\mathcal{S}^n$ is the set of $n \times n$ symmetric matrices. The augmented Lagrangian of its dual problem is defined as

$$L_\mu(U, y, S, Z) = -b^\top y + \langle U, A_V - \mathcal{A}^*(y) - S - Z\rangle + \frac{1}{2\mu}\|A_V - \mathcal{A}^*(y) - S - Z\|_{\mathsf{F}}^2,$$

with $\mu > 0$. Then $U$ is computed by performing the following steps:

$$y^{(t+1)} = \text{argmin}_{y \in \mathcal{R}^{n+1}} L_\mu(U, y, S, Z), \tag{75}$$
$$Z^{(t+1)} = \text{argmin}_{Z \in \mathcal{S}^n} L_\mu(U, y, S, Z), Z \geq 0, \tag{76}$$
$$S^{(t+1)} = \text{argmin}_{S \in \mathcal{S}^n} L_\mu(U, y, S, Z), S \succeq 0, \tag{77}$$
$$U^{(t+1)} = \frac{S^{(t+1)} - W^{(t+1)}}{\mu} \tag{78}$$

with $W^{(t+1)} = A_V - \mathcal{A}^*(y^{(t+1)}) - S^{(t+1)} - Z^{(t+1)}$. $y^{(t+1)}$ can be computed directly from the first-order optimality conditions for equation 75. The solutions of equation 76 and equation 77 are, respectively, $Z^{(t+1)} = \max(W^{(t+1)}, 0)$ and $S^{(t+1)} = Q_+ \Sigma_+ Q_+^\top$, where the diagonal matrix $\Sigma_+$ contains all positive eigenvalues of $W^{(t+1)}$ and $Q_+$ contains the corresponding eigenvectors.

## E  ADMM for solving (16) and (17)

We demonstrate the ADMM algorithm proposed by Kulis et al. (2007); Tepper et al. (2018) using equation 16 as a sample case. The solver for equation 17 follows a similar approach. The augmented Lagrangian of equation 16 is defined as

$$L(Y_U^\top Y_U, \mu, \lambda) = -\langle A_V, Y_U \rangle + \frac{1}{2}\|Y_U^\top Y_U \mathbf{1} - \mathbf{1}\|_2^2 - \lambda^\top(Y_U^\top Y_U \mathbf{1} - \mathbf{1}) - \mu(\mathrm{Tr}(Y_U^\top Y_U) - r_U) + \frac{1}{2}(\mathrm{Tr}(Y_U^\top Y_U) - r_U)^2,$$

where $\mu$ and $\lambda$ are Lagrange multipliers. Then $Y_U$ is computed by performing the following steps:

$$Y_U^{(t+1)} = \mathrm{argmin}_{Y_U \geq 0} L(Y_U, \mu^{(t)}, \lambda^{(t)}),$$
$$\lambda^{(t+1)} = \lambda^{(t)} - Y_U^\top Y_U \mathbf{1} + \mathbf{1},$$
$$\mu^{(t+1)} = \mu^{(t)} - \mathrm{Tr}(Y_U^\top Y_U) + r_U,$$

where the minimization for solving $Y_U^{(t+1)}$ is computed by the L-BFGS-B algorithm introduced by Byrd et al. (1995) with constraints $0 \leq Y_{ij} \leq 1$.

