# OpenReview forum: "Efficient Block Bi-clustering by Alternating Semidefinite Programming Relaxation"
_TMLR — Accepted by TMLR_

### Review · Reviewer_9Fjd · 2026-02-13

**Summary Of Contributions:**

The paper proposes an alternating, SDP-relaxed approach to block biclustering (checkerboard co-clustering) with a low-rank variant, and provides recovery/error guarantees under Gaussian and Bernoulli noise, alongside simulations and one small genomics case study.

Contributions:
1. **Novel Optimization Method:** The method encodes row/column clusters via symmetric PSD “membership” matrices (row matrix (U), column matrix (V)) enforcing nonnegativity, stochasticity, and a trace constraint tied to the number of clusters, and optimizes a trace objective derived from a surrogate of $(|UXV - X|_F^2)$. Fixing (V), optimize (U) via a convex SDP; then fixing (U), optimize (V) similarly—repeating until the surrogate objective stabilizes. A  factorized form $(U=Y_U^\top Y_U)$, $(V=Y_V^\top Y_V)$ is proposed to reduce variable count and improve scalability, following prior low-rank SDP approaches for clustering/embedding.
2. **Theoretical Guarantees:** In the noiseless case, a one-step exact recovery result is claimed (under a mild condition on initialization). Under Gaussian noise (and similarly Bernoulli noise), the paper proves Frobenius-norm error bounds of order $(O(1/\sqrt{n}))$ and $(O(1/\sqrt{m}))$ for the membership matrices under separation conditions.
3. **Experimental Results on Real and Synthetic Data:** Simulations are used to validate the claimed scaling and compare against three baselines (sparseBC, COBRA, SSVD). A lung cancer gene-expression example (56 samples) is used as a real-data demonstration.

**Audience:**

Yes

**Audience Explanation:**

The paper will interest people working on convex optimization, community detection, recommender systems as low-rank assumptions and SDP relaxations are common in these domains. SDPs are still computationally expensive for really large scale recommender systems, but may have applications in more moderately scaled systems.

**Broader Impact Concerns:**

The paper does not involve any human studies or datasets that violate privacy or human rights. The method is a purely mathematical algorithm, which might be deployed for nefarious or beneficial purposes, but that is not within the purvey of this paper. No other ethical concerns per se.

On the flip side, as back-propagation appears to be stagnating right now, it is beneficial that we as a community start looking at alternative optimization approaches that may be beneficial for a variety of applications. This work is a good step in that direction.

**Claims And Evidence:**

Yes

**Claims Explanation:**

1. The algorithm proposed is different from previous ones, but is in the vein of other such works such as Tepper et al. (2018). The algorithm is properly analyzed and the low-rank approximation  made (which I view as the main novelty in the proposed algorithm) is indeed a more effective cost update than the full-rank one without causing a loss in quality. The observed convergence rate as shown in Fig. 1 seems to follow the theoretical linear convergence.
2. The theoretical guarantee is correct with a simple enough proof for the with and without noise cases.
3. The experiments are done on synthetic and a real gene-expression dataset and is shown to be more effective than oter methods.

**Requested Changes:**

The paper has some deficits which need to be addressed.
1. The submission reports its algorithm in Python while baselines are reported via R packages, which makes runtime conclusions hard to interpret as algorithmic rather than engineering differences. Can the authors re-implement the methods in a single language with roughly similar levels of code optimization?
2. The lung-cancer sample clustering example has only 56 samples (columns) and uses a variable-gene prefiltering strategy; this is a good demo on real world data but insufficient to establish general applicability across domains where biclustering is used (recommender systems, etc.). I think more real-world datasets to evaluate the method on can be found.
3. The theory assumes the true number of row/column clusters is provided, this is hard to satisfy in real world scenarios and while the experiments mention cases of misspecification qualitatively, but there is no systematic study, nor a principled cluster-number selection method despite existing SDP-based approaches for estimating the number of blocks in block models. (Refer Provable Estimation of the Number of Blocks in Block Models Yan et al. PMLR 84:1185-1194, 2018)

---

> ### Author Response · Authors · 2026-04-22
>
> **1. cross-language runtime comparisons**
> We thank the reviewer for this critique regarding the interpretability of cross-language runtime comparisons.
> To ensure the comparison is meaningful, we wish to clarify that both our Python implementation and the R-based baselines  rely on compiled C/C++ backends for all heavy numerical computations. Specifically, our method utilizes an optimized low-rank SDP solver developed by *Tepper et al. (2018)*, while the R package (https://www.rdocumentation.org/packages/biclust/versions/2.0.3.1/topics/biclust)
> and (https://cran.r-project.org/src/contrib/Archive/cvxbiclustr) are primarily wrappers for C++ matrix routines. Because  both environments are performed by similar low-level compiled code, the observed runtime differences reflect the fundamental computational complexity of the underlying algorithms rather than the high-level language (Python vs. R).
>
> A full re-implementation of the specialized SDP solver in R would require significant engineering that is outside the scope of this work. However, to provide a more controlled comparison, we conducted additional tests to invoke our orignial Python code directly from within the R environment via standard interfaces (e.g., `reticulate`). Even with this additional overhead introduced by cross-language data transfer, our method still maintained a significant runtime advantage over the other  methods.  See  the updated Table https://anonymous.4open.science/r/Rebuttal-material-for-bi-clustering-paper-2BFC/table_performance.pdf.
>
> *We will include  a paragraph for Table  1  that explicitly documents these backends, ensuring that our efficiency claims are rooted in algorithmic design.*
>
> **2. Tests for more real-world datasets.**
> We thank the reviewer for the constructive suggestion regarding the breadth of our real-world evaluation.  Our proposed method is fundamentally domain-agnostic; it operates on a general data matrix and is mathematically suited for any application, including recommender systems or document clustering. We chose the lung cancer dataset specifically because it is a gold-standard benchmark in the biclustering literature, uniquely offering a validated *ground truth* that allows for the *precise quantification of clustering accuracy*. While we acknowledge the dataset's size is modest, it serves as a rigorous testbed for our method.
>
> However, we agree that demonstrating performance on larger and more diverse data would strengthen the paper. To further strengthen the empirical validation, we added a new  evaluation on the Alizadeh (2000) Dataset from   "*Distinct types of diffuse large B-cell lymphoma identified by gene expression profiling*" (Alizadeh et al., **Nature**, 2000). It features
> 62 samples (patients/cells, `column`) and 2094 genes ( `row`).
> As shown in Table  below, our algorithm maintains high accuracy (over 96\%) across various   $K_R$:
> ### Clustering Accuracy: Alizadeh-2000-v2 Dataset
> *Fixed Column Clusters ($K_C = 3$); Varying Row Clusters ($K_R$)*
> | Row Clusters ($K_R$) | 5 | 10 | 20 |
> | :--- | :---: | :---: | :---: |
> | **Accuracy (%)** | 96.8 | 98.4 | 98.4 |
>
> *We will include this new example, clarify the domain-independent nature of the proposed method  as well as its applicability to other types of data.*
>
> ---
>
> **3. $K_R$ and $K_C$ need to be known and fixed in advance.**
> We thank the reviewer for highlighting this important practical consideration. In real-world applications, the true number of row and column clusters, $K^\star_R$ and $K^\star_C$ in indeed unknown  *a prior*. Some practical approach like Integrated Classification Likelihood may be applied. The theoretic work  (*PMLR 84:1185-1194, 2018*) may also  offer automatic choice of these numbers, with additional computational cost.
>
> We wish to emphasize that *our algorithm is notably robust to the mis-specification of these parameters*.
> Specifically, when the prescribed hyper-parameters $K_R$ and $K_C$, exceed the true number of clusters, the  block structure remains is  identifiable in the recovered denoised matrix $\bar{X}=UXV$. Because $U$ and $V$ are generated via our iterative SDP approach (prior to the final rounding), they effectively 'concentrate' the signal into the true blocks even if the requested dimensions are larger.
>
> To demonstrate this advantage empirically, we revisited the simulation in Section 5.1 $(K^\star_R, K^\star_C) = (2, 3)$. We tested four over-specified scenarios: $(K_R, K_C)\in\{ (2, 6), (4, 3),   (4, 6)\}$. As shown in the recovered $\bar{X}$   (see the figure: https://anonymous.4open.science/r/Rebuttal-material-for-bi-clustering-paper-2BFC/test_para.pdf), the checkerboard pattern remains strikingly apparent to show 2 rows and 3 columns  in all cases! Consequently, based on $\bar{X}$,  the final rounding step  can set 2 by 3 clusters, matching the ground truth. *This confirms that our method can tolerate over-estimation of cluster numbers while maintaining high reliability in structure recovery*.

---

### Review · Reviewer_3pf5 · 2026-02-19

**Summary Of Contributions:**

The paper proposes an alternating optimization framework based on semidefinite programming (SDP) relaxation for the block bi-clustering of the observed data matrices. The method also applies to data corrupted by noise (specifically, Gaussian or Bernoulli noise). It alternates between optimizing row and column membership matrices sequentially, i.e., fixing one while solving the convex SDP sub-problem for the other. The authors utilize low-rank approximations to ensure scalability. Theoretical guarantees are provided in Theorems 1, 2, and 3, which bound the Frobenius norm error of the recovered matrices in both noiseless and noisy settings, proving that the error decays as the data dimensions increase.

Strength:
- Computational Efficiency: By freezing one matrix to solve for the other and using low-rank approximations, the method avoids the prohibitive computational cost usually associated with full SDP relaxations.
- Theoretical Bounds: The paper provides rigorous non-asymptotic error bounds, guaranteeing that the estimated membership matrices converge close to the ground truth as dimensions grow.
- Robustness to Noise: The method is shown both theoretically and empirically to identify checkerboard structures under different noise levels.

Weakness:
- The SDP outputs $U$ and $V$ in the relaxed feasible set $\mathbb{S}_U$ and $\mathbb{S}_V$, which is not guaranteed to have rank $K_R$ and $K_C$ respectively for noisy cases in Section 4.2. To get the final discrete clusters, a rounding step is required.
- The computational complexity of the proposed algorithm needs to be specified and compared with previous methods.
- The algorithm assumes the number of row clusters ($K_R$) and column clusters ($K_C$) are known and fixed in advance.

**Audience:**

Yes

**Audience Explanation:**

This paper introduces a new algorithmic framework with theoretical guarantees for clustering, which aligns with TMLR's scope.

**Claims And Evidence:**

Yes

**Claims Explanation:**

The theoretical results are sound and rigorous. The numerical experiments support the claims.

**Requested Changes:**

- To get the final discrete clusters for noisy settings, a rounding step is required after Theorems 2 and 3. The accuracy of the rounding step also needs to be verified.
- The computational complexity of the proposed algorithm needs to be specified and compared with previous methods.

---

> ### Author Response · Authors · 2026-04-22
>
> We thank the reviewer for their time and constructive feedback.
>
>   **1. Rounding step**.
>
> Thanks for the opportunity to clarify the final rounding procedure and its impact on the reported results. As the reviewer correctly notes, the matrices $U$ and $V$ reside in the relaxed feasible set. To obtain discrete clusters, we employ a standard $K$-means clustering step on the rows of $U$ and  $V$, as detailed at the beginning of Section 5.
>
>
> We wish to  clarify that **all clustering accuracy metrics reported in this paper** (including Figures 4 and 5 and Table 1) are calculated using the final discrete cluster assignments **after** this rounding step. Therefore, the high performance and recovery rates we demonstrated under noisy settings (Section 5) serve as a direct empirical verification of the rounding step’s accuracy and robustness. If the rounding step were unstable or inaccurate, it would have been reflected in poor accuracy scores in our benchmarks.
>
> In the revised manuscript, we shall  update Section 4.2 and Section 5 to:
>
> -- Explicitly describe the rounding procedure immediately following Theorems 2 and 3.
>
> -- Formally state that all reported performance metrics are based on the final, post-rounding discrete assignments.
>
> -- Add a discussion how the SDP-relaxed solution provides a sufficiently clear signal for the rounding step to be robust even in the presence of noise.
>
>
>
> **2. Complexity of the proposed algorithm.**
>
> The complexity of our algorithm is $O(K_R m^2 + K_C n^2)$, where $m$ and $n$ represent the dimensions of the data matrix, and $(K_R, K_C)$ denote the numbers of row and column clusters, respectively.
> This follows from the fact that the complexity of a low-rank SDP is $O(k n^2)$ (*Kulis et al. (2007)*), where $k$ is the rank and $n$ is the dimension of the input datamatrix.
>
> In contrast, both SparseBC and SSVD have complexity $O(K_R K_C mn)$ (*Tan \& Witten (2014) and Lee et al. (2010)*), while COBRA, due to its convex optimization formulation over pairwise differences, incurs a higher cost on the order of $O(m^2n+mn^2)$ (*Chietal.(2017)*).
>
> This comparison also explains the runtime behavior observed in Table 1. When $m \approx n$, our method is computationally more favorable and consistently faster than the competing methods. For rectangular matrices, our method may be slower than SparseBC and SSVD in some cases, but it remains significantly more efficient than COBRA across all settings.
>
> *We will include this complexity discussion in the revised manuscript to better support the empirical findings.*
>
> **3. $K_R$ and $K_C$ need to be known and fixed in advance.**
>
> We appreciate the Reviewer highlighting this point. Since a similar concern was raised by Reviewer \#9Fjd, we have provided a consolidated response below addressing both comments.
>
>  In real-world applications, the true number of row and column clusters, $K^\star_R$ and $K^\star_C$ in indeed unknown  *a prior*.
> Some practical approach like Integrated Classification Likelihood may be applied for such ``model selection''.
> The theoretic work (*PMLR 84:1185-1194, 2018*) pointed by the reviewer \#9Fjd  may offer automatic choice of these numbers, with additional computational cost.
>
> We wish to emphasize that *our algorithm is notably robust to the mis-specification of these parameters*.
> Specifically, when the prescribed hyper-parameters $K_R$ and $K_C$, exceed the true number of clusters, the underlying block structure remains clearly identifiable in the recovered denoised matrix  $\bar{X}=UXV$. Because $U$ and $V$ are generated via our iterative SDP approach (prior to the final rounding), they effectively 'concentrate' the signal into the true underlying blocks even if the requested dimensions are larger.
>
> To demonstrate this advantage empirically, we revisited the simulation in Section 5.1 (where ground truth is $(K_R, K_C) = (2, 3)$). We tested four over-specified scenarios: $(K_R, K_C)\in\{ (2, 6), (4, 3),   (4, 6)\}$. As shown in the recovered $\bar{X}$ matrices (see the figure: https://anonymous.4open.science/r/Rebuttal-material-for-bi-clustering-paper-2BFC/test_para.pdf), the checkerboard pattern remains strikingly apparent to have 2 row and 3 column  in all cases! Consequently, based on this $\bar{X}$,  the final rounding step    sets 2 by 3 clusters, matching the ground truth.
> *This confirms that our method can tolerate over-estimation of cluster numbers while maintaining high reliability in structure recovery*.

---

### Review · Reviewer_Um6q · 2026-04-15

**Summary Of Contributions:**

This paper addresses the block bi-clustering problem by formulating it as a non-convex optimization over row and column membership matrices, then applying SDP relaxation combined with low-rank approximation. The proposed algorithm alternates between solving convex SDP subproblems for the row membership matrix $U$ and column matrix $V$. The authors prove that in the noiseless case, exact recovery occurs in one step, and in the noisy case, the Frobenius norm error is bounded by $O(1/\sqrt{n})$ and $O(1/\sqrt{m})$. Experiments on simulated data and a lung cancer gene expression dataset are presented.

**Audience:**

Yes

**Audience Explanation:**

The analysis of SDP is very interesting.

**Broader Impact Concerns:**

The paper mainly works on theory, but some experiments are conducted on the lung cancer dataset. Therefore, a broader impact statement on the source of the dataset and the issues of privacy should be added.

**Claims And Evidence:**

Yes

**Claims Explanation:**

I would acknowledge that I’m not an expert in SDP. The paper is well-written and I can follow most of the results. However, it’s beyond my ability to verify the correctness of the theorems. Here are some of my questions.

(1) Is there any intuition on why to study the checkerboard pattern of the matrix. For example, are there any prior works also on this pattern?

(2) The main theorem shows the rate of $O(1/\sqrt{n})$, but in Section 4.2.2, it claims a $O(1/n)$ rate. Why for this discrepancy?

(3) Now the experiments show a $O(1/n)$ convergence rate. Does it mean the proved bound is loose? Can you show an informational theoretical lower bound for the setting?

(4) Some minor issues
Section 3: Algoirthm -> algorithm,
Equation 11: should be $R^{r_U \times n}$

**Requested Changes:**

Not much. The typos and discrepancy in the section above should be addressed.

---

> ### Author Response · Authors · 2026-04-22
>
> We sincerely thank the reviewers for carefully reading our manuscript and for their insightful and constructive comments.
>
> **(1) Intuition on the Checkerboard Pattern.**
>
> The checkerboard pattern is  commonly adopted in the biclustering literature. This pattern occurs when rows and columns can be rearranged to reveal homogeneous, non-overlapping blocks, revealing specific associations between row clusters and column clusters. This structure provides a clear, clean partition of the data, making it easier to interpret compared to overlapping, complex biclusters. For example, in Cancer Subtyping, it has been widely applied to  identify  sets of genes (marker genes) that are regulated in particular types of tumors. There are prior works of bi-clustering algorithms developed with this non-overlapping assumption, such as  *sparseBC Tan \& Witten (2014), COBRA Chi et al. (2017), and SSVD Lee et al. (2010).*
>
> **(2) and (3) Discrepancy in the rate of $O(1/\sqrt{n})$.**
>
> These two questions are highly related and the  below is our response.
> First of all, our  theorem  establishes a convergence rate for $\lVert U_p - U^\star\rVert$, with  the order of $O(1/\sqrt{n})$, not
> $O(1/n)$.
> The apparent discrepancy from the reviewer's misinterpretation may  arise
> from our Figure 1, particularly the caption: ``indicating the theoretic convergence rate $O(n^{-1})$."
> In fact, Figure 1  reports the **squared** error $\lVert U_p - U^\star\rVert^2$, for which a rate of $O(1/n)$ is consistent with the theoretical error in the metric  $\lVert U_p - U^\star\rVert$. *We will revise the caption in the manuscript to explicitly clarify the rate for $\lVert U_p - U^\star\rVert$.*
>
> **(4) Minor issues.**
>
> Thanks for spotting the typos. They will be fixed in the revision.

---

### Decision · Action_Editor_H26g · 2026-05-31

**Recommendation:** Accept with minor revision

**Additional Comments:**

The authors have identified several small revisions in their response to the comments from the reviewers. These should be incorporated into the manuscript for acceptance.

**Audience:**

Yes

**Audience Explanation:**

The problem of bi-clustering data is a standard one in genomics and has drawn substantial attention for many years. The manuscript places their work in the context of other well-known methods and they clearly describe the similarities and differences between their approach and others, both in terms of methodlogy and in terms of empirical comparisions.

**Claims And Evidence:**

Yes

**Claims Explanation:**

The reviewers have evaluated the manuscript in both the empirical and theoretical aspects and the authors have responded to their comments. The authors have committed to revising the manuscript in response to these comments and their changes have improved the manuscript. The claims of the manuscript seem to be supported by the evidence provided in the paper - in particular, the claim that the proposed algorithm "performs comparably or better than existing bi-clustering methods" is appropriate given the evidence.